# Mitochondrial translocation of TFEB regulates complex I and inflammation

Chiara Calabrese[1,2], Hendrik Nolte[1,2], Melissa R Pitman [3], Raja Ganesan [3], Philipp Lampe[1], Raymond Laboy [1,2], Roberto Ripa[2], Julia Fischer [1,4], Ruhi Polara [3], Sameer Kumar Panda [3,5], Sandhya Chipurupalli[3], Saray Gutierrez[1], Daniel Thomas [6,7], Stuart M Pitson [3], Adam Antebi [1,2,6 ✉] & Nirmal Robinson [1,3 ✉]

## Abstract

TFEB is a master regulator of autophagy, lysosome biogenesis, mitochondrial metabolism, and immunity that works primarily through transcription controlled by cytosol-to-nuclear translocation. Emerging data indicate additional regulatory interactions at the surface of organelles such as lysosomes. Here we show that TFEB has a non-transcriptional role in mitochondria, regulating the electron transport chain complex I to down-modulate inflammation. Proteomics analysis reveals extensive TFEB co-immunoprecipitation with several mitochondrial proteins, whose interactions are disrupted upon infection with *S.* Typhimurium. High resolution confocal microscopy and biochemistry confirms TFEB localization in the mitochondrial matrix. TFEB translocation depends on a conserved N-terminal TOMM20-binding motif and is enhanced by mTOR inhibition. Within the mitochondria, TFEB and protease LONP1 antagonistically co-regulate complex I, reactive oxygen species and the inflammatory response. Consequently, during infection, lack of TFEB specifically in the mitochondria exacerbates the expression of pro-inflammatory cytokines, contributing to innate immune pathogenesis.

**Keywords** TFEB; Mitochondria; Metabolism; LONP1; Salmonella
**Subject Categories** Metabolism; Microbiology, Virology & Host Pathogen Interaction; Organelles

## Introduction

The transcription factor EB (TFEB) belongs to the family of basic helix-loop-helix leucine zipper (bHLH-ZIP) containing proteins and is a key regulator of autophagy, lysosomal biogenesis and cellular metabolism (Lapierre et al, 2013; Napolitano and Ballabio, 2016; Settembre and Ballabio, 2011) with increasingly recognized roles in the regulation of innate immunity (Irazoqui, 2020). TFEB nuclear activity is dependent on the mammalian target of rapamycin (mTOR), which under nutrient-rich conditions, phosphorylates TFEB at Ser 122, Ser142 and Ser211 facilitating binding to 14-3-3 adaptor proteins, and preventing its nuclear translocation and transcriptional activity (Martina et al, 2012). Conversely, mTOR inhibition during lysosomal stress and starvation results in reduced TFEB phosphorylation, permitting its nuclear translocation and transcriptional function (Settembre et al, 2012b). TFEB has also been shown to co-localize with mTOR on the surface of lysosomes, in a RAG GTPase-dependent manner (Settembre et al, 2012b) serving as a lysosomal "sensor" in addition to its established role as a lysosomal transcriptional "effector". Recently, the structure of TFEB presented by Rag–Ragulator complexes to be phosphorylated by mTOR has been solved (Cui et al, 2023).

Mitochondrial function is maintained via a balance of biogenesis, fission, fusion and specific autophagy (mitophagy). Mitophagy is partially regulated by TFEB (Nezich et al, 2015; Zhang et al, 2016), especially in response to mitochondrial stress such as that induced by electron transport chain inhibition and induces TFEB-dependent expression of mitophagy receptors NDP52 and Optineurin (OPTN). Beta islet cell specific knockout of TFEB abrogated high-fat diet-induced mitophagy with increased reactive oxygen species (ROS) production (Park et al, 2022), indicating a critical link between mitochondrial electron transport complex function and TFEB. Moreover, TFEB transcriptionally regulates genes involved in mitochondrial biogenesis and glucose uptake (Mansueto et al, 2017) and loss of TFEB results in accumulation of morphologically abnormal and dysfunctional mitochondria (Mansueto et al, 2017). More recently, TFEB has been reported to induce mitochondrial itaconate production in macrophages, which inhibits bacterial growth (Schuster et al, 2022) and regulates extrusion of mitochondrial contents in hepatocytes exposed to lipopolysaccharide (LPS) (Unuma et al, 2015) but, direct non-transcriptional roles of TFEB on mitochondrial function have not been identified.

[1]Cologne Excellence Cluster on Cellular Stress Responses in Aging-Associated Diseases (CECAD), University of Cologne, Cologne, Germany. [2]Max Planck Institute for Biology of Ageing, Cologne, Germany. [3]Centre for Cancer Biology, University of South Australia and SA Pathology, Adelaide, Australia. [4]Centre for Molecular Medicine Cologne, Cologne, Germany. [5]Department of Experimental Medicine, University of Campania "Luigi Vanvitelli", Naples, Italy. [6]Adelaide Medical School, University of Adelaide, Adelaide, Australia. [7]Precision Medicine Theme, South Australian Health and Medical Research Institute, Adelaide, Australia. ✉E-mail: aantebi@age.mpg.de; Nirmal.Robinson@unisa.edu.au

Effective upregulation of autophagosomes and lysosomes via TFEB is required for xenophagy and the destruction of intracellular pathogens such as *Mycobacteria* (Ouimet et al, 2016)) and *Salmonella* (Schuster et al, 2022), constituting an essential innate immune response. TFEB regulates inflammation and immunity upon infection in macrophages in a transcriptional program conserved in the nematode *Caenorhabditis elegans* (Pastore et al, 2016; Visvikis et al, 2014) with recent data indicating mitochondrial metabolism may contribute to the effector function of TFEB activated macrophages (Schuster et al, 2022).

Here we report that TFEB has a previously unknown mitochondria-specific function. We have established that TFEB translocates to mitochondria via a TOMM20-binding motif forming a complex with LONP1 to regulate electron transport complex I. Our observations show a direct link between mitochondrial ROS production and innate immune defense regulated directly by TFEB.

## Results

### TFEB localizes to the mitochondrial matrix

To identify novel functions of TFEB in an unbiased manner, we performed high affinity mass spectrometry to identify proteins interacting with ectopically expressed TFEB in uninfected cells vs *S.* Typhimurium-infected or torin-1-treated cells. We investigated *S.* Typhimurium infection as we had previously shown this pathogen activates mTOR, prevents autophagy and induces inflammation (Ganesan et al, 2017). Significant interacting partners were determined by two-tailed *t*-test and then corrected for multiple testing by permutation-based FDR estimation (FDR < 0.05) (Appendix Fig. 1A). In total, we identified 79 proteins that showed twofold upregulation in untreated conditions. As expected, 15 core candidates including 14-3-3 proteins (YWHAQ, YWHAB, YWHAG, YWHAE, YWHAZ, YWHAH) which are well established binding partners of TFEB (Xu et al, 2019) were significantly twofold upregulated compared to control in all treatments, validating our immunoprecipitation results (Fig. 1B, Dataset EV1, Appendix Fig. S1A). As expected, (Xu et al, 2019) the intensity of TFEB binding with 14-3-3 proteins were reduced upon torin-1 treatment (Fig. 1B and Appendix S1A).

We then selected proteins that were significantly upregulated at least twofold in any treatment. This set contains 130 proteins indicating a highly dynamic TFEB interactome (Dataset EV1). Strikingly, 28 out of 130 proteins localized to the mitochondria, including ATPase Family AAA Domain Containing 3B (ATAD3B), Leucine Rich Pentatricopeptide Repeat Containing (LRPPRC), Heat Shock Protein Family A (Hsp70) Member 1B (HSPA1B), and Heat Shock Protein Family A (Hsp70) Member 9 (HSPA9) (Fig. 1C,D). The majority of proteins (75 out of 130) were associated with the nucleus (Fig. 1C) including TFEB, signal transducer and activator of transducer and activation of transcription 3 (STAT3), Proteasome 20S Subunit Alpha 4 (PSMA4) and these interactions were mostly maintained in untreated (UT) conditions and upon torin-1 treatment but reduced upon *S.* Typhimurium (ST) infection (Fig. 1D). Taken together, these data suggested a potential localization and function of TFEB in mitochondria.

Consistent with this notion, confocal microscopy indicated that endogenous TFEB colocalized with MitoTracker Red (Fig. 1E). The Manders co-localization coefficient (Dunn et al, 2011) of TFEB to MitoTracker (tM1) and MitoTracker to TFEB (tM2) were very similar as quantified by colocalization imageJ analysis (Fig. 1F), indicating significant overlap. We then established a HeLa cell line in which TFEB was stably depleted by shRNA (shTFEB) (Appendix Figs. S1B,S1C). Cell fractionation of shRNA control (shCTRL) and shTFEB followed by Western blotting showed the presence of TFEB in the mitochondrial fraction from shCTRL but not shTFEB cells (Fig. 1G,H). As previously reported (Settembre et al, 2011b), autophagy was inhibited in TFEB-depleted cells, as the levels of the autophagy marker LC3B is decreased both at the basal level and upon torin-1 treatment (Appendix Fig. S1D). We also isolated mitochondria from HEK293T cells using anti-TOM22 microbeads, which yielded highly enriched mitochondria (TOMM20) devoid of contaminants from endoplasmic reticulum (ER) (PDI), cytoplasm (GAPDH) and nucleus (LAMIN B). Notably, TFEB was found to be enriched in mitochondrial fraction (Fig. 1I) and was not specific to cancer cell lines, as we also detected TFEB in the mitochondrial fraction of primary human monocyte derived macrophages (Appendix Fig. S1E). However, other proteins MITF and TFE3 belonging to the bHLH-ZIP family transcription factors were not found on isolated mitochondria (Appendix Fig. S1F).

Immunofluorescence stimulated emission depletion (STED) microscopy revealed that TFEB was in close proximity with mtDNA but not directly binding to mtDNA (Fig. 2A) suggesting that TFEB is located in the same compartment as mtDNA and the mtDNA staining was confirmed to be present in TOMM20-stained mitochondria (Appendix Fig. S2A). To identify more precisely the localization of TFEB within the mitochondria, we performed protease protection assays on mitochondria isolated from HeLa cells (Fig. 2B,C). In the absence of swelling buffer, trypsin degraded the outer mitochondrial membrane (OMM) protein TOMM20. In the presence of swelling buffer, trypsin degraded the inner mitochondrial membrane (IMM) protein TIM23. By contrast, the matrix protein SLP2 was resistant to trypsin digestion, TFEB was partially degraded suggesting that it could be partially present on the inner membrane but primarily localized within the mitochondrial matrix. However, TFEB was not resistant to trypsin per se, as both TFEB and SLP2 were digested by trypsin when the mitochondria were incubated with triton X-100. We also confirmed that TFEB was localized to the mitochondrial matrix in HEK293T cells (Appendix Figs. S2B,S2C). In addition, immunogold labeling of TFEB followed by transmission electron microscopy (TEM) detected gold particles in contact with the mitochondrial cristae, as well as the nucleus and cytoplasm (Fig. 2D,E).

### Mitochondrial localization signal (MLS) of TFEB enables mitochondrial translocation

The majority of the proteins destined to the mitochondrial matrix are synthesized with matrix-targeting sequences (MTS) (Backes and Herrmann, 2017). We predicted a putative MTS for TFEB using MitoProt II (https://ihg.gsf.de/ihg/mitoprot.html) and generated a TFEB construct in which we deleted the putative MTS (ΔMTS) to assess whether this domain is responsible for its mitochondrial translocation. Cell fractionation performed on HeLa cells

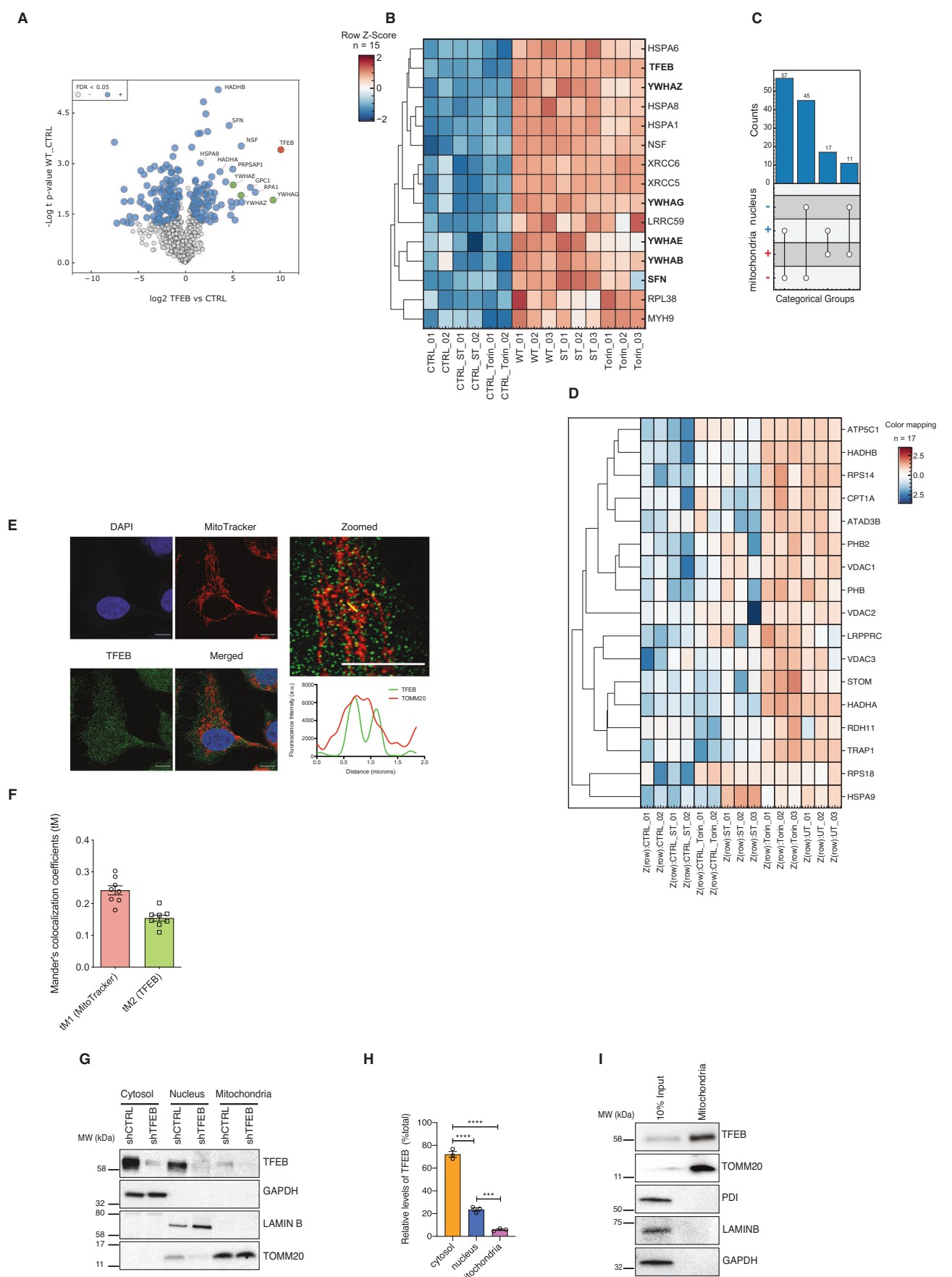

**Figure 1. TFEB localizes to mitochondria.**

(A) Volcano plot of FLAG-immunoprecipitates from cells transfected with WT-TFEB FLAG and an unrelated protein as control. The blue dots, green dots and red dots represent positive interactors, established TFEB interactors and high enrichment of TFEB respectively, indicating the reliability of the proteomics data. (B) Hierarchical clustering using Euclidean distance and complete method for row dendrogram calculations representing the core TFEB interactome. Shown are individual replicates of the WT-TFEB pull down in untreated conditions (UT), upon torin-1 treatment and upon *S.* Typhimurium infection (ST). The CTRL is represented by a pull down of another non-specific-FLAG tagged protein used as negative control for the interactions. Log2 LFQ intensities were Z-Score normalized. 14-3-3 proteins as well as the bait TFEB are highlighted in bold font. (C) Countplot indicating cellular localization (based on Gene Ontology cellular component annotations) of identified potential interaction partners. "+" symbol corresponds to a known localization either in mitochondria or nucleus. "-" symbol corresponds to an unknown localization in the same compartments. Points and lines indicate categorical group and the bar graph indicates the number of proteins matching the categorical group. (D) Hierarchical clustering of TFEB-interacting mitochondrial proteins. Shown are individual replicates similar to (A). (E) HeLa cells stained for mitochondria using MitoTracker red and nucleus. Scale bar = 10 μM. (F) Colocalization of TFEB and MitoTracker from (D) was evaluated in ROIs (25 × 25 pixels) as described in Methods section. Mander's colocalization coefficients using the calculated thresholds (tM) were determined for the red (tM1) (TOMM20) and the green (tM2) (TFEB) channels. Eight cells per group were analyzed. Error bars represent standard error mean (SEM). (G) TFEB expression in subcellular fractions from shCTRL and shTFEB HeLa cells. LAMINB, TOMM20, and GAPDH served as loading controls for the nucleus, mitochondria, and cytoplasm, respectively. Western blot is representative of three independent experiments showing similar results. (H) Quantitation of endogenous TFEB levels in subcellular fractions of HeLa cells. Error bars denote SEM; $n = 3$ biological replicates; One-way ANOVA followed by Tukey's multiple comparison test was conducted (***$p < 0.005$, ****$p < 0.001$). (I) TFEB expression in 10% of total lysate (positive control) and in highly enriched mitochondria isolated from HEK293T cells using TOMM22-magnetic beads. TOMM20 was used to confirm the enrichment of mitochondria. PDI was used to exclude ER contaminations; LAMINB to exclude nuclear contaminations and GAPDH to exclude cytosolic contaminations. Blot shown is a representative image of atleast 3 replicates. Source data are available online for this figure.

expressing an empty FLAG plasmid, FLAG-tagged WT-TFEB or FLAG-tagged ΔMTS-TFEB showed that the ΔMTS-TFEB translocated into the mitochondria, indicating that translocation is independent of the predicted MTS (Appendix Figs. S3A,S3B). We also identified a specific TOMM20 binding consensus motif (VMHYM) in TFEB using MitoFates (http://mitf.cbrc.jp/MitoFates/cgi-bin/top.cgi), which we termed as the mitochondrial localization sequence (MLS). To test whether this motif directed TFEB translocation into mitochondria, we mutated the MLS from the WT-TFEB plasmid (aa 30 from V to K) by site directed mutagenesis to change the charge (Fig. 3A). Cell fractionation analyses on HeLa cells expressing MLS-TFEB revealed that the mutated MLS prevented TFEB translocation to the mitochondria (Fig. 3B,C). Mitochondrial import assays using $^{35}$S radiolabeled TFEB translated *ex vivo* further demonstrated that TFEB was actively imported into mitochondria in a membrane-potential-dependent manner as CCCP treatment significantly reduced TFEB translocation into mitochondria. However, $^{35}$S radiolabeled MLS-TFEB translated ex vivo showed an impairment in mitochondrial translocation, further demonstrating the function of the MLS motif in mitochondrial import (Fig. 3D,E).

TFEB co-localization with TOMM20 was also tested by confocal microscopy and quantified for both WT-TFEB FLAG and the mutant MLS-TFEB FLAG. We observed significantly increased co-localization of WT-TFEB FLAG with TOMM20 compared to the MLS-TFEB FLAG. However, no difference in nuclear localization was detected between the WT-TFEB and the mutant MLS-TFEB (Appendix Figs. S3C,S3D). We found that the TOMM20 binding motif is conserved in higher mammals, but not present in zebrafish or *C. elegans* (Fig. 3F). In line with this, the *C. elegans* TFEB orthologue, HLH-30, was not found to reside in mitochondria (Fig. S3E).

Because lysosomal biogenesis is a crucial transcriptional function of TFEB (Settembre et al, 2011a) we asked whether the MLS mutant preserved TFEB transcriptional activity by analyzing the mRNA of its lysosomal target genes. Reduced target gene expression in shTFEB was rescued when the cells were reconstituted with either WT-TFEB or MLS-TFEB mutant plasmids (Appendix Fig. S3F), indicating that TFEB transcriptional activity is uncompromised in the MLS-TFEB transfected cells.

## Mitochondrial translocation of TFEB is mTOR-dependent

As TFEB nuclear translocation is mTOR-dependent (Settembre et al, 2012a), we investigated whether TFEB mitochondrial localization also shows mTOR regulation. Inhibition of mTOR using torin-1 in HeLa cells increased the global expression of TFEB and its nuclear translocation as TFEB is also known to autoregulate itself (Settembre et al, 2013), but also promoted TFEB mitochondrial translocation (Fig. 3G,H). Immunofluorescence staining of HeLa cells showed that endogenous TFEB and TOMM20 colocalization increased upon torin-1 treatment (Fig. 3I,J).

This finding is consistent with our mass spectrometry analysis (Fig. 1C), in which we observed a positive interaction between TFEB and a subset of mitochondrial proteins upon torin-1-treatment. TFEB subcellular localization is known to be mediated by mTOR-dependent phosphorylation of TFEB residues Ser122, Ser142, and Ser211 (Settembre and Ballabio, 2011). We next investigated the phosphorylation status of TFEB in mitochondria (M) to the total cell lysate (C) in HeLa cells by mass spectrometry. This analysis revealed that the pool of TFEB present in the mitochondria was not phosphorylated on the identified phosphorylation sites (Ser122, Ser332, and Ser334) (Fig. 3K). The phosphorylation of TFEB on these Ser residues was lost upon torin-1 treatment (Fig. 3K). However, Ser467 was the only phosphorylated residue to be detected in the mitochondria pool upon torin-1 treatment. Ser467 phosphorylation is known to be responsible for TFEB cytosolic stabilization, as it prevents TFEB nuclear translocation independently of mTORC1 (Palmieri et al, 2017). Furthermore, upon torin-1 treatment, TOMM20 co-immunoprecipitated with endogenous TFEB (Fig. 3L). These data suggest that inhibition of mTOR allows the translocation of the dephosphorylated pool of TFEB into mitochondria, which is dependent on the identified MLS motif.

## Loss of TFEB disrupts mitochondrial morphology and function

Next, we studied whether mitochondrial morphology and function were affected upon TFEB depletion. (Settembre et al, 2011b) We observed that the mitochondrial network was altered, as

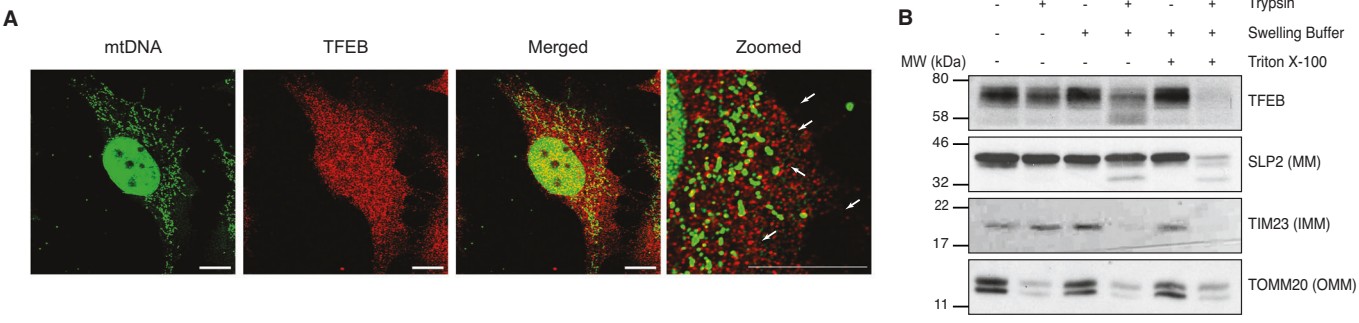

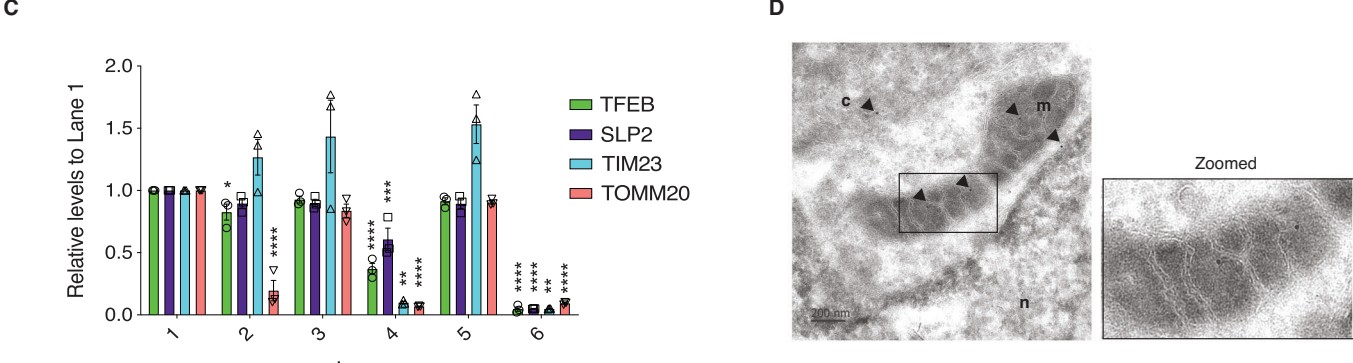

**Figure 2. TFEB translocates into the mitochondrial matrix in a membrane-potential-dependent manner.**

(A) HeLa cells were stained for mitochondrial DNA (mtDNA) using the fluorescent DNA-specific dye PicoGreen and endogenous TFEB. Arrows indicate co-localization. Scale bar = 10 μM. (B) Protease protection assay on purified mitochondria isolated from HeLa cells using 50 μg trypsin in the presence or absence of swelling buffer. Samples were analyzed for the outer mitochondrial membrane (OMM) protein TOMM20, the inner mitochondrial membrane (IMM) protein TIM23, and the mitochondrial matrix (MM) protein SLP2. SLP2 and endogenous TFEB were exposed to trypsin after mitochondria were lysed with Triton X-100. Western blot is representative of two independent experiments showing similar results. (C) Densitometric analysis of protein levels in (B). Band intensities were normalized to the first lane (untreated mitochondria) of the respective protein. Error bars denote SEM; $n = 3$ biological replicates; One-way ANOVA followed by Tukey's multiple comparison test was conducted (*$p < 0.05$; **$p < 0.01$; ***$p < 0.005$; ****$p < 0.001$). (D) Immunogold endogenous-TFEB labeling in mitochondria of HeLa cells. Arrows indicate immunogold particles detected in the nucleus (n), cytosol (c) and mitochondria (m). Scale bar = 200 nm. (E) Synthetic dataset of observed gold particles deposited on organelle-based compartments. Gold particles were counted using stereology method, applied on 20 images. $G_o$ values indicate the number of gold particles observed; $G_e$ values indicate the number of gold particles expected. Source data are available online for this figure.

mitochondria clustered asymmetrically in the perinuclear region in TFEB-depleted cells (Fig. 4A,B). The mitochondrial network was rescued with ectopic expression of WT-TFEB but not when complemented with the MLS-TFEB mutant (Fig. 4C). TFEB mitochondrial localization and altered mitochondrial morphology in TFEB-depleted cells, prompted us to investigate the impact of TFEB on mitochondrial respiration (Fig. 4D). Seahorse analysis showed increased ATP-linked respiration, and Maximal respiration in shTFEB, which dropped to baseline when cells were reconstituted with WT-TFEB, but not with the MLS-TFEB mutant (Fig. 4E). Consistently, increased ATP-linked respiration and

Maximal respiration was observed also in HEK293T cells when transiently transfected with siRNA against TFEB compared to siCTRL (Appendix Fig. S4A). Taken together, these data indicate that TFEB has a mitochondria-specific function in regulating mitochondrial oxidative phosphorylation.

To better dissect the mitochondria specific functions of TFEB, we generated plasmids which express various mutant forms of TFEB. TFEB not able to translocate to the nucleus (ΔNLS-TFEB) was created by removing the nuclear localization signal (NLS) motif (in which the basic arginine residues R245–R248 were mutated to alanine). TFEB that exclusively localizes in the nucleus was

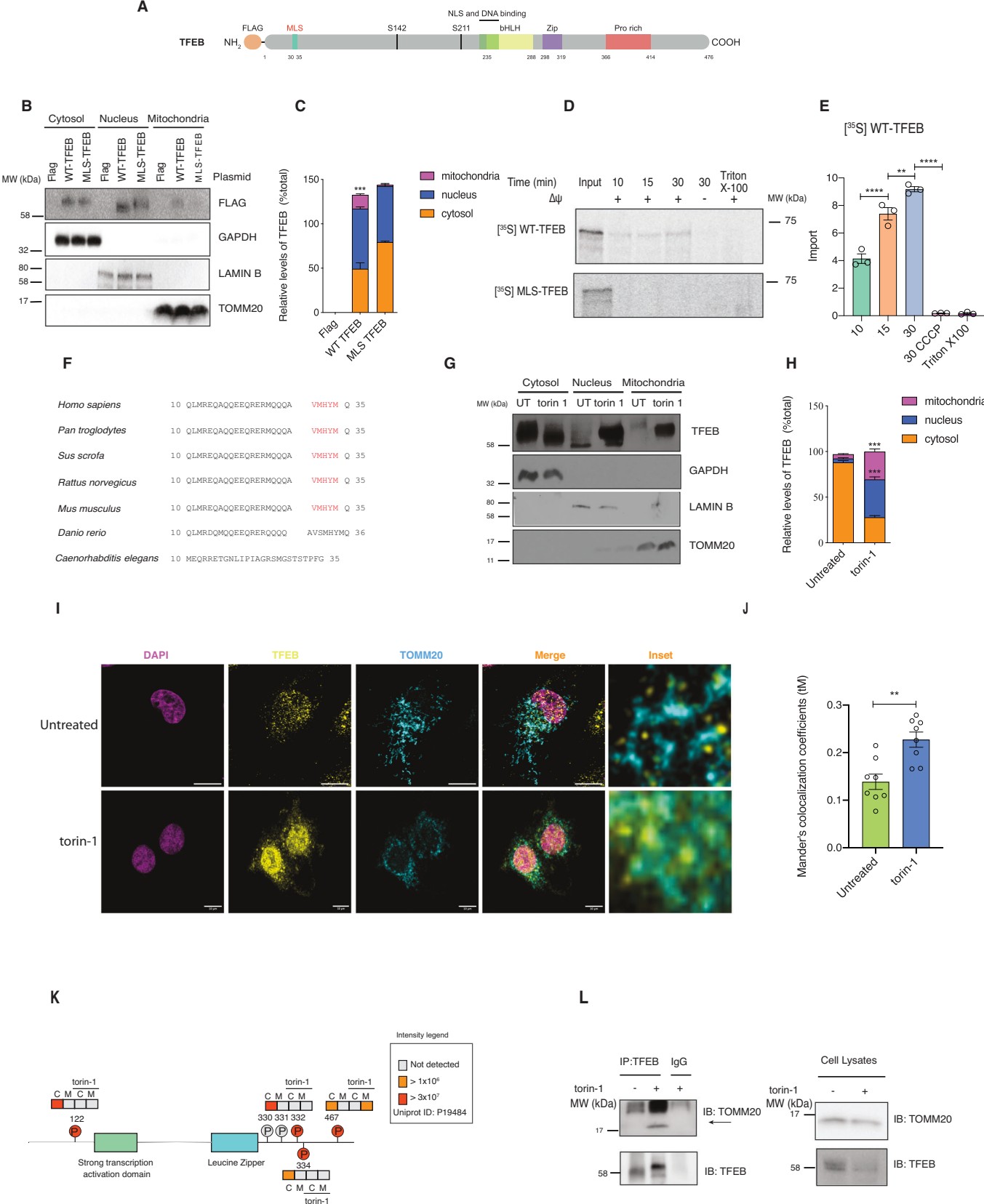

**Figure 3. TFEB import into mitochondria depends on a mitochondria localization sequence (MLS) and mTOR.**

(A) Graphical representation of the WT-TFEB FLAG plasmid showing the predicted MLS domain. (B) Analysis of subcellular fractions isolated from HeLa cells expressing FLAG, WT-TFEB- FLAG or MLS-TFEB-FLAG. LAMINB, TOMM20, and GAPDH served as controls for nucleus, mitochondria, and cytoplasm, respectively. Western blot is representative of three independent experiments showing similar results. (C) Relative abundance of TFEB in the various fractions of the immunoblot shown in (B). Error bars denote SEM; $n = 3$ biological replicates; unpaired *t*-test was conducted, and statistical significance is denoted as ***$p < 0.005$. (D) Import assay using in vitro synthesized [$^{35}$S]WT-TFEB and [$^{35}$S]MLS-TFEB performed on isolated mitochondria. Samples were treated with trypsin and assay was performed in the presence (+) or absence (−) of mitochondrial membrane potential (Δψ) at different time points. Mitochondria lysed with Triton X-100 were loaded as control. (E) Graph represents quantification of the in vitro [$^{35}$S]WT-TFEB import assay shown in (C). Error bars denote SEM; $n = 3$ biological replicates; One-way ANOVA followed by Tukey's multiple comparison test was conducted and statistical significance denoted as **$p < 0.01$; ***$p < 0.005$; ****$p < 0.001$. (F) Conservation of the MLS sequence (red). (G) Endogenous TFEB in subcellular fractions isolated from HeLa cells in untreated conditions (UT) and upon 1 h torin-1 treatment. LAMIN B, TOMM20, and GAPDH served as controls for nucleus, mitochondria, and cytoplasm, respectively. Western blot is representative of three independent experiments showing similar results. (H) Relative abundance of TFEB in the various fractions of the immunoblot shown in (G). Error bars denote SEM; $n = 3$ biological replicates; Unpaired *t*-test was conducted and statistical significance is represented as ***$p < 0.005$. (I) Confocal image of HeLa cells untreated and treated with torin-1 stained for TFEB and TOMM20. Scale bar = 10 μM. (J) Colocalization of TFEB and TOMM20 from (I) was evaluated in ROIs (25x25 pixels) as described in the "Methods" section. Mander's colocalization coefficients using the calculated thresholds (tM) were determined for the cyan (TOMM20) and yellow (TFEB) channels. Eight cells per group were analyzed. Graph shows Mander's colocalization coefficients using the calculated thresholds (tM) for TFEB channel in untreated and torin-1 treated HeLa cells. Error bars represent SEM; unpaired *t*-test was conducted, and statistical significance represented as **$p < 0.01$. (K) TFEB structure showing phosphorylation sites in untreated and torin-1-treated cells identified and quantified by LC-MS/MS. Colors indicate intensities in the cytosol (C) or mitochondria (M). (L) Endogenous TFEB was immunoprecipitated (IP) from untreated and torin-1 treated HeLa cells and immunoblotted (IB) for TOMM20 and TFEB; Isotype specific IgG was used as negative control. The immunoprecipitations were performed twice independently showing similar results. Source data are available online for this figure.

generated by mutating the previously reported serine residues (S142A-S211A) to alanine (Napolitano et al, 2018; Roczniak-Ferguson et al, 2012). Both constructs were additionally mutated in the identified MLS motif to prevent their mitochondrial translocation. In addition, we generated tagging the MTS of the mitochondrial protein superoxide dismutase 2 (SOD2) with TFEB (MTS-TFEB) which strongly enriched TFEB in mitochondria (Appendix Fig. S4B). Subcellular fractionation (Appendix Fig. S4C) and confocal microscopy (Fig. S4E) confirmed the cytosolic localization of the ΔNLS-TFEB mutant and the nuclear localization of the S142A/S211A TFEB mutant. Mitochondrial enrichment of MTS-TFEB was also confirmed by cell fractionation (Appendix Fig. S4D) and confocal microscopy (Appendix Fig. S4E).

We next complemented shTFEB with the generated TFEB plasmids and evaluated their ability to restore mitochondrial function. Interestingly, Basal respiration, ATP-linked respiration and Maximal respiration were rescued in shTFEB cells when complemented with WT-TFEB and MTS-TFEB, but not when complemented with the ΔNLS-TFEB or S142A/S211A-TFEB (Appendix Figs. S4F, S4G and S4H). These data indicate that mitochondrial respiration is regulated by mitochondria localized TFEB. Transcriptional regulation of lysosomal genes was restored when shTFEB cells were complemented with WT-TFEB or S142A/S211A-TFEB. However, MTS-TFEB and ΔNLS-TFEB which do not translocate to nucleus did not restore the transcriptional function of TFEB (Fig. 4F). Also, we investigated if the transcription of mitochondrial genes is TFEB-dependent. None of the mitochondrial gene expressions we studied were significantly altered upon the loss of TFEB and complementation with WT or the mutant TFEB plasmids did not cause any significant changes (Appendix Fig. S4I).

## Mitochondrial complex I assembly is altered in TFEB-depleted cells

To understand how mitochondrial TFEB regulates mitochondrial functions, we performed label-free proteomics on enriched mitochondria from shCTRL and shTFEB to determine the altered expression of mitochondrial proteins. 1D enrichment of Gene

Ontology (GO) terms was performed along the $\log_2$ shTFEB/shCTRL ratio to identify up- or down-regulated proteins annotated with the same GO term. Proteins annotated as *mitochondrial respiratory chain complex I* were upregulated in shTFEB compared to shCTRL (Fig. 5A and Appendix Figs. S5A and S5B). Complex I subunits were plotted based on their localization and modules (Wirth et al, 2016) to see whether subunits pertaining to a specific complex I module were altered. We found an overall increase in the expression of complex I subunits in TFEB-depleted cells, unlinked to a specific complex I module (Appendix Fig. S5C).

We next investigated complex I by blue native-polyacrylamide gel electrophoresis (BN-PAGE) and found an increased complex I in shTFEB compared to shCTRL (Fig. 5B). We used the mitochondria isolated from the same cells to investigate complex I activity by in-gel activity assay in response to the complex I substrates NADH and Nitro Blue Tetrazolium (NBT). Complex I activity was also modestly increased in shTFEB compared to shCTRL (Fig. 5C). Increased complex I activity was validated using complex I enzyme activity microplate assay (Fig. 5D). Further, expression of several complex I subunits such as NDUFS5, NDUFA8 and NDUFB10, but not NDUFA9 were higher in the mitochondria isolated from shTFEB than shCTRL (Fig. 5E,F). We also determined the levels of subunits of other ETC complexes (Appendix Figs. S5D,S5E) and found that expression of proteins belonging to complex II were also affected in TFEB-depleted cells. SDHA was increased in TFEB-depleted cells, whereas SHDB was decreased. Given that SDHA is also part of the TCA cycle, changes in this protein could also be due to a different metabolic profile of TFEB-depleted cells. However, expression of complex III (UQCRC2), complex IV (COX2) and complex V (ATP5a and ATP5b) associated proteins were not altered in enriched mitochondria of TFEB-depleted cells compared to shCTRL (Appendix Figs. S5D,S5E).

To test whether the increase in these specific complex I subunits was dependent on mitochondrial TFEB, we studied the expression of NDUFB10 and NDUFA8 in shTFEB cells complemented with WT-TFEB or MLS-TFEB mutant (Fig. 5G,H). Expression of complex I subunits was reduced when shTFEB cells were reconstituted with WT-TFEB but not with the MLS-TFEB mutant.

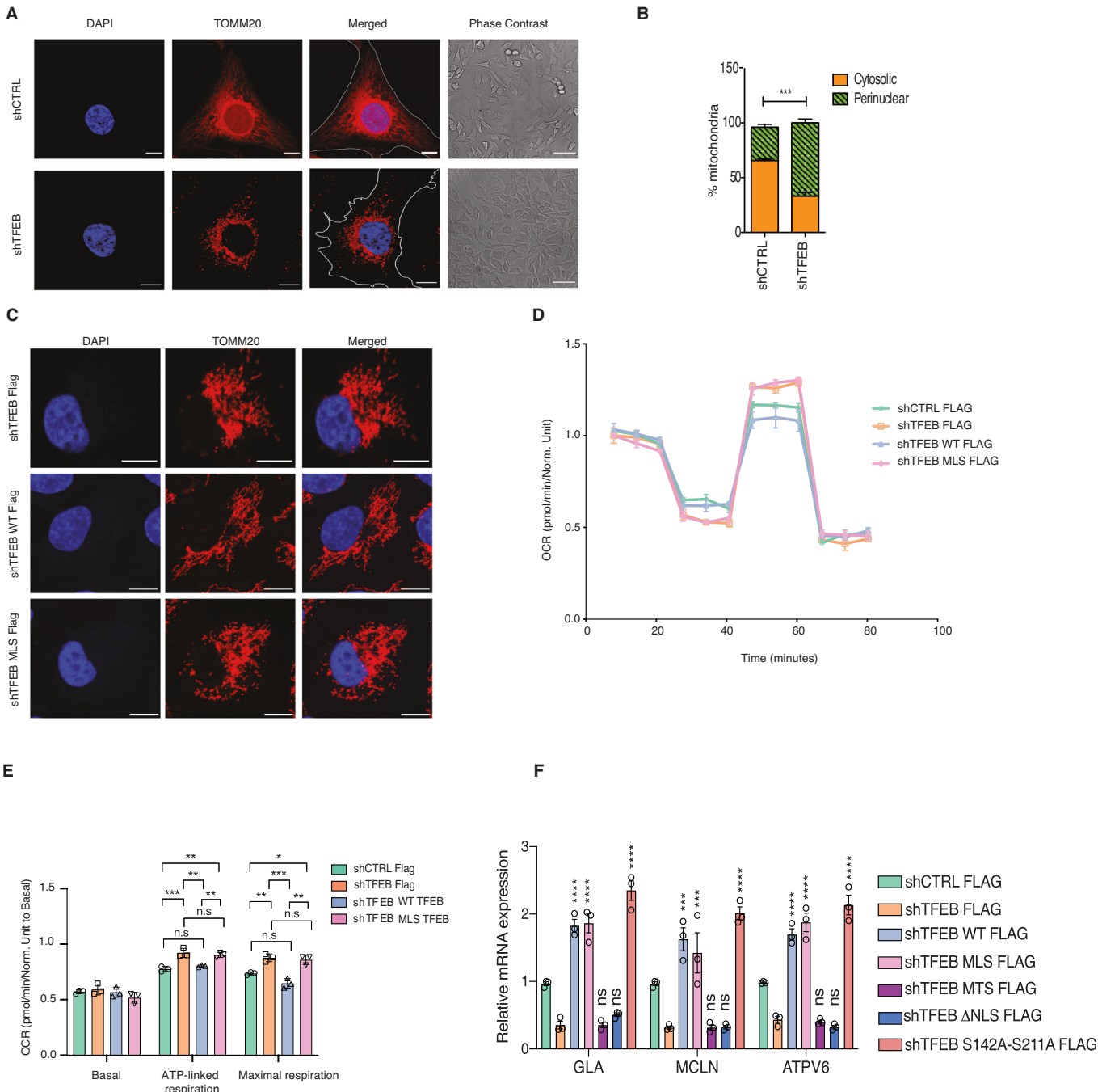

**Figure 4. Loss of TFEB dysregulates mitochondrial architecture and function.**

(A) TOMM20 and nucleus staining in shCTRL or shTFEB HeLa cells. Scale bar = 10 μm. The boundaries of the cell are shown with a marking. Phase contrast images of the two cell lines are also shown to compare the size of the cells. (B) Percentage of perinuclear and cytosolic mitochondria clustering. $n = 50$ cells each. Error bars represent SEM; unpaired t-test was conducted, and statistical significance represent as ***$p < 0.005$. (C) TOMM20 and nucleus staining in shTFEB cells transfected with FLAG, WT-TFEB- FLAG or MLS-TFEB-FLAG. Scale bar = 10 μm. (D) OXPHOS profile normalized to Basal respiration of shCTRL and shTFEB cells transfected with empty FLAG, WT-TFEB and MLS-TFEB plasmids. Data represents 3 biological replicates; error bars denote SEM. (E) Basal, ATP-linked respiration and Maximal respiration normalized to Basal respiration in shCTRL and shTFEB cells transfected with FLAG, WT-TFEB- FLAG or MLS-TFEB-FLAG. Shown are mean ± SEM, $n = 3$ biological replicates; one-way ANOVA followed by Tukey's multiple comparison analysis was done, and statistical significance represented as *$p < 0.05$; **$p < 0.01$; ***$p < 0.005$; ****$p < 0.001$. (F) mRNA levels of TFEB target genes (relative to HPRT) from shTFEB HeLa cells transfected with FLAG, WT-TFEB- FLAG, MLS-TFEB-FLAG, ΔNLS-TFEB FLAG, S142A/S211A-TFEB FLAG or MTS-TFEB FLAG. Shown are mean ± SEM, $n = 3$ biological replicates. One-way ANOVA followed by Tukey's multiple comparison analysis was done, and statistical significance represented as *$p < 0.05$; **$p < 0.01$; ***$p < 0.005$; ****$p < 0.001$. Source data are available online for this figure.

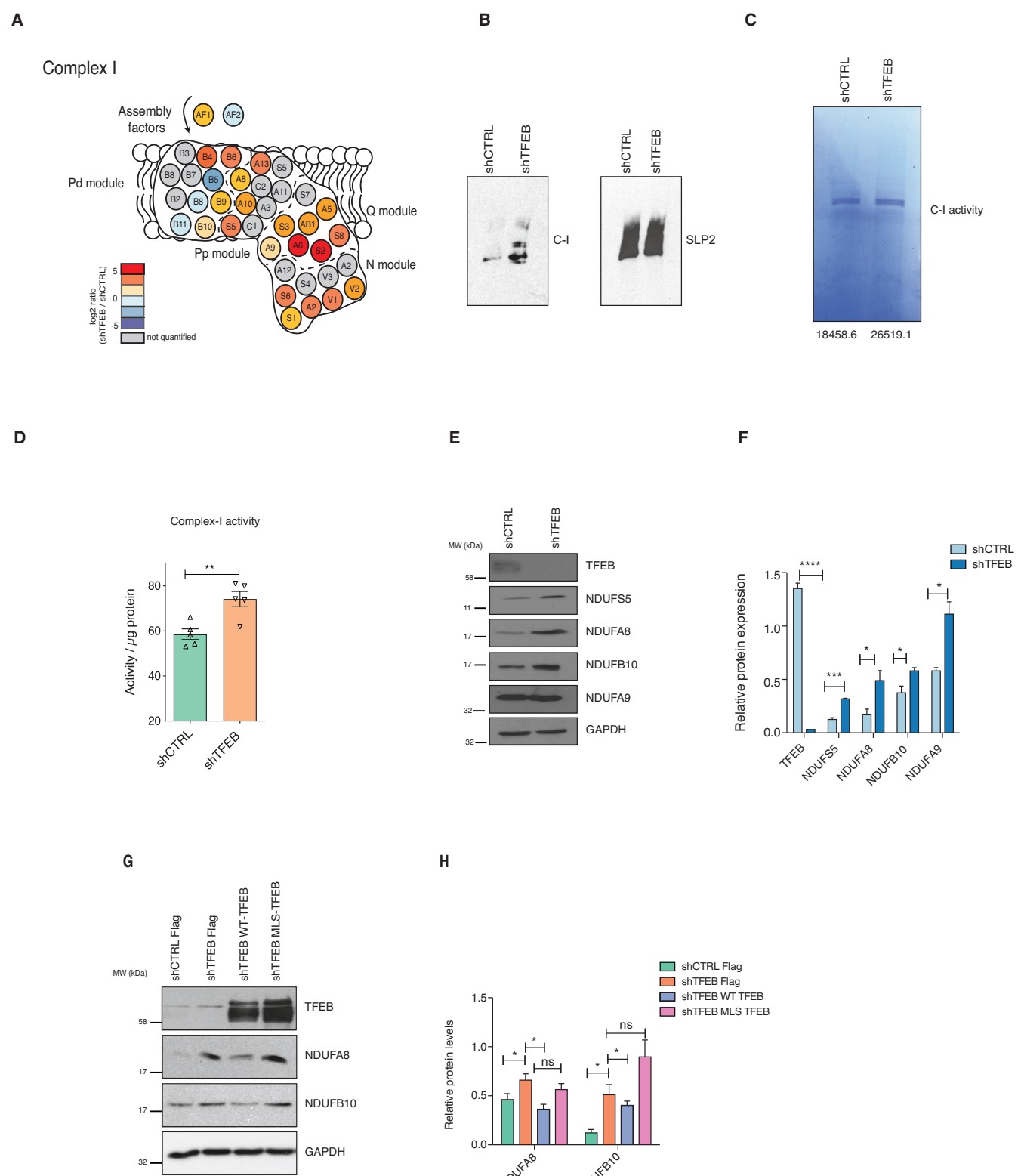

Collectively, these findings suggest that TFEB-regulated complex I function is mitochondria-specific. We also investigated if TFEB depletion altered mitophagy. PINK1 accumulates on the outer mitochondrial membrane when mitochondrial homeostasis is altered and mitophagy is activated, thus used as a hallmark of mitophagy. We observed increased PINK1 colocalization with TOMM20 in TFEB-depleted cells complemented with WT-TFEB, MLS-TFEB, and S142A/S211A TFEB. However, PINK1

Figure 5.  Mitochondrial complex I is altered in TFEB-depleted cells.

(A) Schematic representation of the respiratory chain complex I. Colors indicate the log2 fold change of shTFEB cells compared to control cells. (B) Blue Native Gel Electrophoresis showing complex I assembly in mitochondria enriched from shCTRL and shTFEB cells. Complex I assembly is representative of three independent experiments showing similar results. SLP2 was used as loading control. (C) In-gel complex I activity after incubation with NADH and NBT in mitochondria enriched from shCTRL and shTFEB cells. Numbers represent the intensity of the bands quantified. Complex I activity is representative of two independent experiments showing similar results. (D) Complex I activity determined using microplate assay in shCTRL and shTFEB HeLa cells. Shown are mean ± SEM, $n = 5$ biological replicates. Unpaired $t$-test was done to determine statistical significance and noted as **$p < 0.01$. (E) TFEB, NDUFS5, NDUFA8, NDUFB10, NDUFA9, and GAPDH expression in shCTRL and shTFEB HeLa cells. Western blot is representative of three independent experiments showing similar results. (F) Mean densitometric analysis of the immunoblot shown in (E). Shown are mean ± SEM, $n = 3$ biological replicates. Unpaired $t$-test was done to determine statistical significance and noted as *$p < 0.05$; ***$p < 0.005$; ****$p < 0.001$. (G) NDUBFA8, NDUFB10, TFEB, and GADPH expression in shCTRL and shTFEB transfected with FLAG, WT-TFEB- FLAG or MLS-TFEB-FLAG. Western blot is representative of three independent experiments showing similar results. (H) Mean densitometric analysis of the immunoblot shown in (G). Shown are mean ± SEM, $n = 3$ biological replicates. Unpaired $t$-test was done to determine statistical significance and noted as *$p < 0.05$. Source data are available online for this figure.

colocalization with TOMM20 was reduced when shTFEB cells were complemented with MTS-TFEB and ΔNLS-TFEB, which can localize in mitochondria but not into nucleus (Appendix Figs. S5F,S5G). This suggests that mitophagy regulated by TFEB is independent of mitochondrial TFEB.

## TFEB and LONP1 co-regulate complex I

Turnover of mitochondrial complexes are regulated by chaperones and mitochondrial proteases (Bohovych et al, 2015). Previous studies have shown that the serine mitochondrial protease LONP1 controls complex I by regulating its subunits (Pinti et al, 2016). Our proteomics approach also identified LONP1 as one of the interactors that immunoprecipitated with TFEB (Dataset EV1) although it was not detected as one of the top hits. Given the effects of TFEB on complex I, we assessed LONP1 expression levels in TFEB-depleted cells. Interestingly, we observed that TFEB knock-down by siRNA increased LONP1 protein levels; conversely, depletion of LONP1 (Fig. 6A,B) or pharmacological inhibition using CDDO (Appendix Figs. S6A,S6B) increased TFEB expression, showing mutual antagonistic regulation. We observed that depleting LONP1 was sufficient to suppress NDUFB10 and NDUFA8 expression in TFEB-depleted cells (Fig. 6A,B), but not of other complex I subunits such as NDUFA9 and NDUFS5 (Fig. 6C and S6C). LONP1 mRNA levels were not significantly changed in cells depleted of TFEB using TFEB-specific siRNA or shRNA (Fig. 6D and Appendix Fig, S6D), indicating a post-transcriptional regulation. No significant differences were found in the mRNA levels of these subunits upon LONP1 knockdown in shCTRL and shTFEB cells (Appendix Fig. S6E). These data indicate that TFEB and LONP1 post-transcriptionally modulate complex I function, by regulating specific subunits known to be upregulated upon LONP1 overexpression (Pinti et al, 2016). Next, we asked if the altered OXPHOS in shTFEB is dependent on LONP1 by silencing LONP1. Interestingly, depleting LONP1 was sufficient to restore normal levels of ATP-linked and maximal respiration in TFEB-depleted cells (Fig. 6E,F). Thus, TFEB and LONP1 cooperatively modulate mitochondrial oxidative phosphorylation.

These data prompted us to investigate whether TFEB and LONP1 form a complex to regulate complex I. We observed that LONP1 immunoprecipitated with endogenous TFEB in HeLa cells (Fig. 6G). Ectopically expressed LONP1-HA also co-immunoprecipitated with TFEB-FLAG in HEK293T cells (Fig. 6H). TFEB-FLAG expression was reduced in the presence of LONP1-HA when compared to TFEB-FLAG expression as control, further indicating a reciprocal regulation of these two proteins (Fig. 6H).

Finally, we purified recombinant LONP1-HA and identified a direct interaction when the protein was incubated with TFEB-FLAG beads (Fig. 6I). The purified proteins (LONP1-HA and TFEB-FLAG) were free of contaminants as assessed by SyproRuby staining after SDS-PAGE (Appendix Fig. S6F). Taken together these data demonstrate that TFEB-LONP1 complex regulates complex I, and therefore mitochondrial function.

## Mitochondrial TFEB regulates inflammation upon S. Typhimurium-infection

Previously, we had shown that *Salmonella* Typhimurium prevents autophagy, disrupts mitochondria (Hos et al, 2017), induces inflammation and triggers phenotypes reminiscent of TFEB loss. Therefore, we surmised that TFEB may be dysregulated during *S.* Typhimurium infection. We first asked whether TFEB transloca-tion is affected upon *S.* Typhimurium infection. Immunofluores-cence microscopy showed that *S.* Typhimurium prevented TFEB nuclear translocation, however, inhibition of mTOR using torin-1 enabled nuclear translocation as expected (Appendix Figs. S7A,S7B). We next evaluated TFEB mitochondrial localization upon *S.* Typhimurium infection. Cell fractionation analyses revealed that TFEB accumulated in the cytosol, however, it was absent from the mitochondria (Fig. 7A,B). These findings are consistent with our data indicating that upon *S.* Typhimurium infection, interactions between TFEB and mitochondrial proteins are lost (Fig. 1E) but preserved with 14-3-3 proteins (Appendix Fig. S1B).

TFEB has been shown to transcriptionally regulate cytokine expression during infection (Visvikis et al, 2014); thus, we next asked whether TFEB cytosolic localization affects the inflammatory outcome during *S.* Typhimurium infection. Depletion of TFEB using shRNA resulted in higher IL-6 (Fig. 7C) and IL-8 (Fig. 7D) secretion compared to control cells in which localization of endogenous TFEB was limited to the cytosol during *S.* Typhimur-ium infection (Appendix Figs. S7A,S7B). Consistently, mRNA levels of IL-6, IL-8 and IL-1α were significantly up-regulated in TFEB-depleted cells (Appendix Fig. S7C).

As TFEB-depleted cells showed increased inflammation (Fig. 7C,D and Appendix S7C), we investigated whether this behavior is dependent on TFEB mitochondrial localization. Although, *S.* Typhimurium prevented TFEB mitochondrial trans-location, ectopic overexpression of WT-TFEB resulted in TFEB mitochondrial localization in shCTRL and shTFEB even in the presence of the bacteria (Appendix Fig. S7D). Therefore, transiently transfected shCTRL and shTFEB cells with WT-TFEB or MLS-

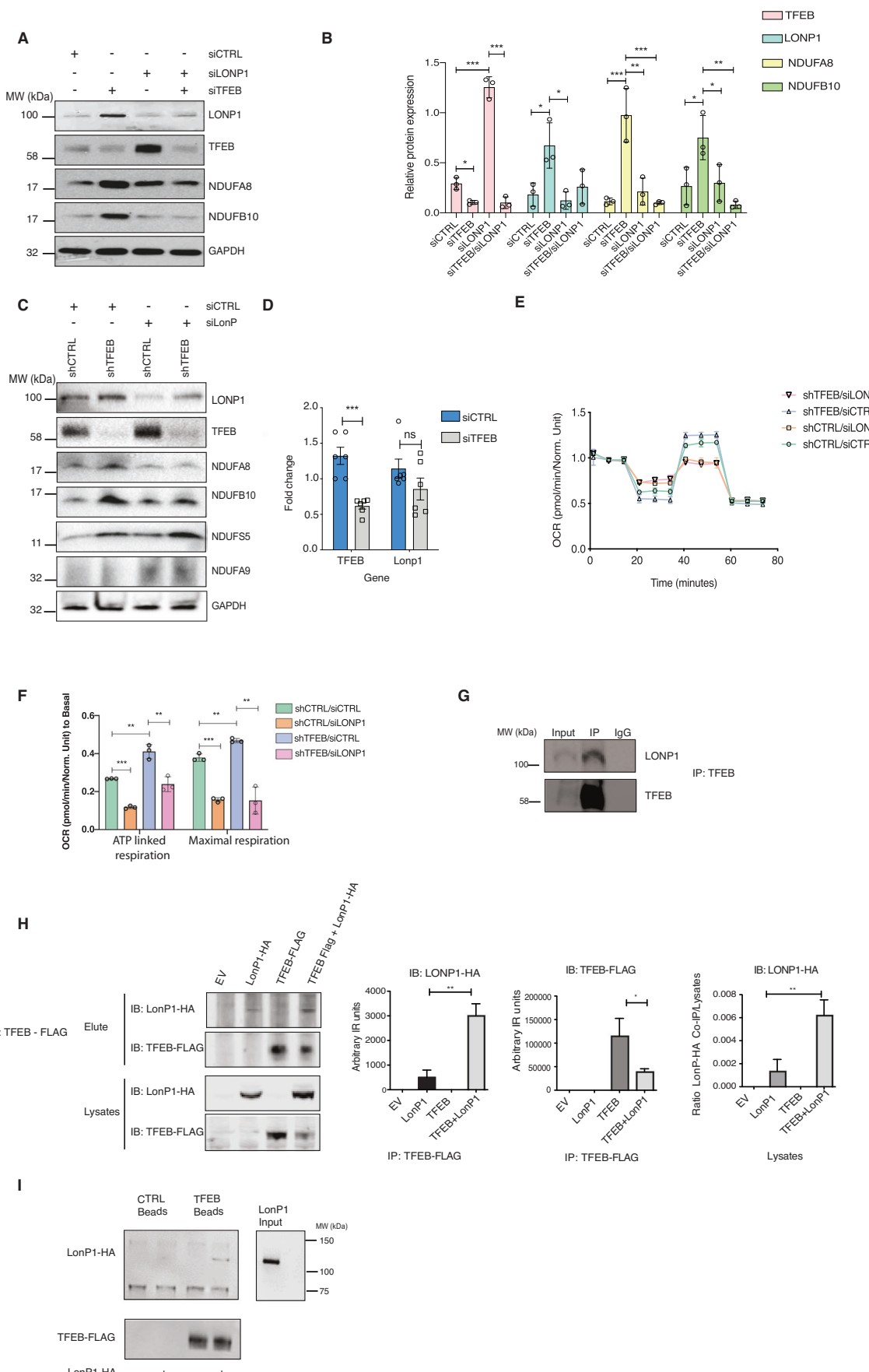

Figure 6.   TFEB and LONP1 co-regulate complex I.

(A) TFEB, LONP1, NDUFA8, NDUFB10, and GAPDH expression in HeLa cells transfected with control siRNA (siCTRL), siRNA against LONP1 (siLONP1), TFEB specific siRNA (siTFEB) and siRNA against both LONP1 and TFEB. Western blot is representative of three independent experiments showing similar results. (B) Mean densitometric analysis of the immunoblot from (A). Shown are mean ± SEM, $n = 3$ biological replicates. Unpaired $t$-test was done to determine statistical significance and represented as *$p < 0.05$; **$p < 0.01$; ***$p < 0.005$; ****$p < 0.001$. (C) LONP1, TFEB, NDUFA8, NDUFB10, NDUFS5, NDUFA9, and GAPDH protein expression in shCTRL and shTFEB HeLa cells transfected with siCTRL and siLONP1. Western blot is representative of three independent experiments showing similar results. (D) Relative mRNA expression of LONP1 and TFEB (relative to RPL13A) in HeLa cells transfected with siCTRL and siLONP1. Shown are mean ± SEM, $n = 6$ biological replicates. Unpaired $t$-test was done to determine statistical significance and represented as ***$p < 0.005$. (E) OXPHOS profile of shCTRL and shTFEB cells transfected with siCTRL and siLONP1. Shown are mean ± SEM, $n = 3$ biological replicates. (F) ATP-linked respiration and Maximal respiration rates normalized to Basal respiration in shCTRL and shTFEB HeLa cells transfected with siCTRL or siLONP1. Shown are mean ± SEM, $n = 3$ biological replicates.. One-way ANOVA test followed by Tukey's multiple comparison was done to determine statistical significance and represented as **$p < 0.01$; ***$p < 0.005$. (G) Endogenous TFEB was immunoprecipitated and immunoblotted for LONP1 and TFEB in HeLa cells. Western blot is representative of two independent experiments showing similar results. IgG was used as negative control. (H) TFEB-Flag was pulled down from cells transfected with Empty vector (EV), LONP1-HA, TFEB-FLAG and TFEB-FLAG + LONP1-HA and immunoblotted for LONP1-HA and TFEB-FLAG to confirm TFEB-LONP1 interaction. EV was used as a negative control. Quantifications of the immunoblot are also shown. The experiment was performed in biological triplicates ($n = 3$). Unpaired $t$-test was done to determine statistical significance and represented as *$p < 0.05$; **$p < 0.01$. (I) LONP1 was purified and incubated with control beads as negative control and with TFEB- FLAG-tagged beads. The experiment was performed in duplicate showing similar results. Source data are available online for this figure.

TFEB mutant were infected with *S*. Typhimurium for 24 h and the supernatants analyzed for IL-6. WT-TFEB but not MLS-TFEB mutant expression reduced IL-6 expression in shTFEB (Fig. 7E), suggesting that TFEB-mediated regulation of inflammation depends also on its mitochondrial localization. Cytokine expression was also analyzed in shTFEB cells complemented with nuclear (S142A/S211A), cytosolic (ΔNLS TFEB), mitochondrial (MTS) TFEB. Nuclear TFEB and cytosolic TFEB did not reduce the *S*. Typhimurium-induced IL-6 and IL-8. However, similar to WT-TFEB, MTS-TFEB restored the induction of IL-6 and IL-8. (Appendix Fig. S7E).

As mitochondrial Reactive Oxygen Species (mtROS) is one of the key factors that regulate inflammation (Naik and Dixit, 2011), we tested whether this could represent one of the causes of increased inflammation observed in TFEB-depleted cells. mtROS produced upon *S*. Typhimurium infection were analyzed by flow cytometry and found to be increased in TFEB-depleted cells compared to control cells and reduced when cells were complemented with the WT-TFEB plasmid (Fig. 7F). Treatment of infected shCTRL and shTFEB cells with the mitochondria-targeted ROS-scavenger, Mitotempo, attenuated the expression of inflammatory cytokines (Fig. 7G), which further confirmed that mtROS was a key mediator of inflammation in the absence of TFEB. We next asked if silencing *LONP1* could reduce inflammation during infection and found that IL-6 production was dampened upon *S*. Typhimurium infection by silencing *LONP1* (Fig. 7H) or pharmacologically inhibiting LONP1 (Appendix Fig. S7F), or upon inhibition of complex I using Rotenone (Appendix Fig. S7G). Consistently, mtROS production was also dampened upon the loss of *LONP1* (Fig. 7I) or its inhibition using CDDO (Fig. S7H). These data suggest that loss of TFEB enhances mtROS-mediated inflammation upon *S*. Typhimurium infection, which is dependent on LONP1-regulated mitochondrial complex I.

## Discussion

The transcriptional role of TFEB in activating several physiological functions and pathologies is well established (Lapierre et al, 2013; Napolitano and Ballabio, 2016); however, to our knowledge a non-nuclear function is unprecedented. Our efforts to understand the cellular defense mechanisms targeted by *S*. Typhimurium led us to

identify a novel mitochondrial role for TFEB. We have deciphered mTOR-dependent translocation of TFEB into mitochondria where it regulates mitochondrial complex I by forming a complex with LONP1 (Fig. 7J).

Starvation and lysosome storage disorder induced TFEB relocation are the most widely studied, but other stimuli such as lipopolysaccharide (LPS) and pathogens like *Staphylococcus aureus* have also been shown to promote TFEB nuclear translocation. We show that *S*. Typhimurium prevents TFEB translocation into the nucleus, consistent with our previous report that *S*. Typhimurium infection only transiently activates autophagy (Ganesan et al, 2017). During *S. aureus* infection, nuclear localization of TFEB is associated with the transcription of pro-inflammatory cytokines (Visvikis et al, 2014). Though *S*. Typhimurium infection prevented TFEB nuclear translocation, TFEB depletion further increased the secretion of pro-inflammatory cytokines underscoring a non-nuclear function of TFEB and inflammation regulated by other factors independent of TFEB.

Through multiple lines of evidence, we have demonstrated that TFEB localizes in mitochondria and critically regulates mitochondrial function, with broad implications for its role in health and disease. We show that TFEB resides mostly in the mitochondrial matrix through immunoprecipitation-MS proteomics, confocal microscopy, STED microscopy, immunogold labeling EM, mitochondrial fractionation, mitochondrial pulldown, protease protection and cell free import assays. In addition, we were able to show the presence of the endogenous TFEB in different cell lines (HeLa and HEK), and in primary human monocyte derived macrophages.

Other transcription factors such as tumor protein P53 (p53), signal transducer and activator of transcription 3 (STAT3), and Nuclear factor kappa-light-chain-enhancer of activated B cells (NF-κB) that regulate inflammatory pathways also localize in the mitochondria (Szczepanek et al, 2012), and in some cases bind to the mtDNA and induce mitochondrial gene transcription. More recently, it has also been reported that the Microphthalmia Transcription Factor (MITF) which also belongs to the same family of basic helix-loop-helix leucine zipper transcription factors as TFEB, interacts with Pyruvate Dehydrogenase (PDH) in the mitochondria modulating its activity, which is crucial in the regulation of the development and the activity of the mast cells (Sharkia et al, 2017).

The majority of the proteins destined to the mitochondrial matrix are synthesized with amino terminal pre-sequence or

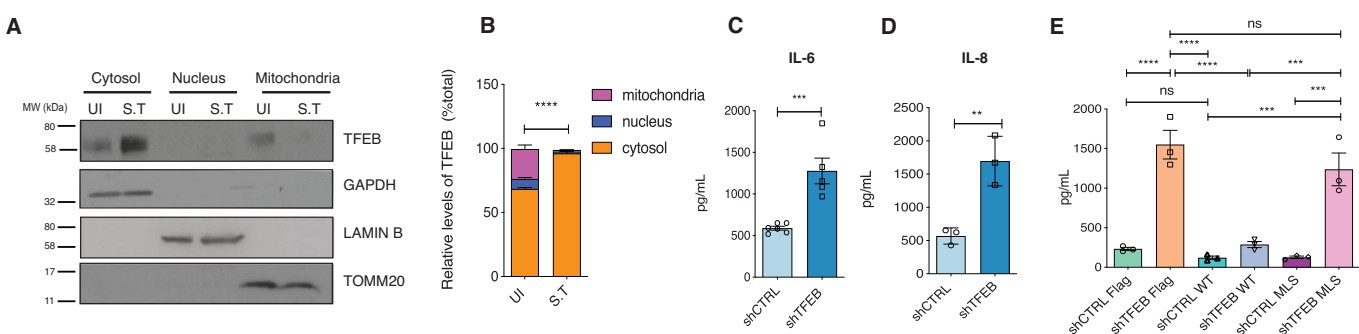

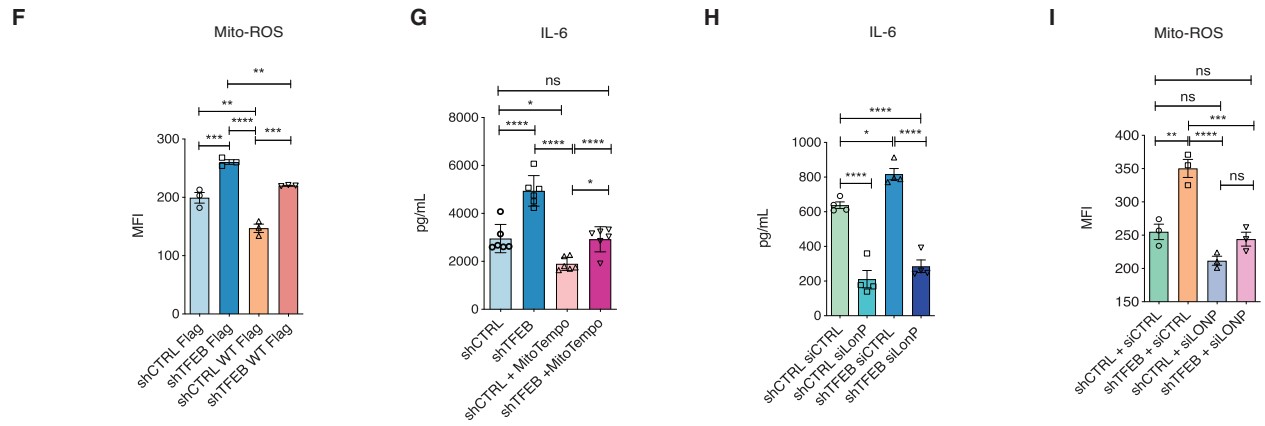

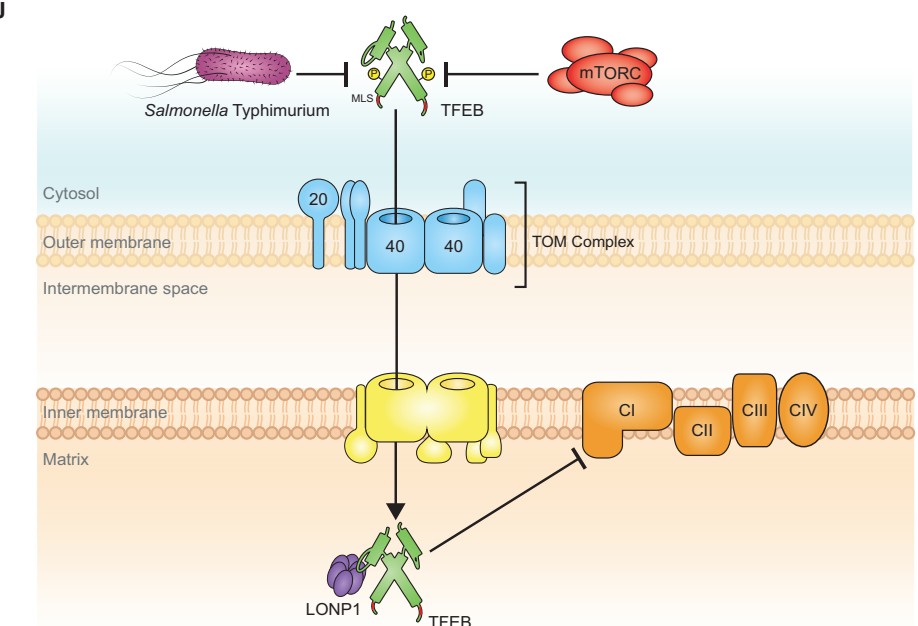

**Figure 7. Mitochondrial TFEB regulates inflammation.**

(A) Endogenous TFEB subcellular localization upon 24 h of S. Typhimurium infection (MOI 100) in HeLa cells. LAMINB, TOMM20, and GAPDH served as controls for nucleus, mitochondria, and cytosol, respectively. Western blot is representative of three independent experiments showing similar results. (B) Densitometric quantitation of immunoblots depicting the subcellular endogenous TFEB localization in uninfected (UI) and S. Typhimurium-infected (MOI = 100) cells from (A). Shown are mean ± SEM, $n = 3$ biological replicates. Unpaired $t$-test was done to determine statistical significance and represented as ****$p < 0.001$. (C, D) (C) IL-6 and (D) IL-8 in the supernatants of 24 h S. Typhimurium-infected shCTRL and shTFEB HeLa cells. Shown are mean ± SEM, $n = 6$ biological replicates for IL-6 (C) and 3 biological replicates for IL-8 (D). Shown are mean ± SEM, $n = 3$ biological replicates. Unpaired t-test was done to determine statistical significance and represented as **$p < 0.01$; ***$p < 0.005$. (E) IL-6 in supernatants of shCTRL and shTFEB cells transfected with FLAG, WT-TFEB-FLAG or MLS-TFEB-FLAG plasmids and infected for 24 h with S. Typhimurium (MOI 100). Shown are mean ± SEM, $n = 3$ biological replicates. Shown are mean ± SEM, $n = 3$ biological replicates. One-way ANOVA followed by Tukey's multiple comparison analysis was performed to determine statistical significance and represented as *$p < 0.05$; **$p < 0.01$; ***$p < 0.005$; ****$p < 0.001$. (F) Flow cytometric analysis of Mitochondrial ROS (mtROS) production was estimated on MitoSOX-stained shCTRL and shTFEB HeLa cells transfected with empty FLAG and WT-TFEB-FLAG and the Mean Fluorescence Intensity (MFI) was plotted; mean ± SEM, $n = 3$ biological replicates. Unpaired $t$-test was done to determine statistical significance and represented as **$p < 0.01$; ***$p < 0.005$; ****$p < 0.001$. (G) IL-6 secretion in supernatants of shCTRL and shTFEB cells infected with S. Typhimurium (MOI 100) for 24 h and treated with MitoTempo 2 h prior infection and during infection. Shown are mean ± SEM, $n = 3$ biological replicates with $n = 3$ technical replicates each. One-way ANOVA followed by Tukey's multiple comparison analysis was performed to determine statistical significance and represented as *$p < 0.05$; ****$p < 0.001$. (H) IL-6 expression in supernatants of shCTRL and shTFEB HeLa cells transfected with siCTRL and siLONP1 and infected with S. Typhimurium (MOI 100) for 24 h. Shown are mean ± SEM, $n = 3$ technical replicates. The experiment was repeated twice with similar results. One-way ANOVA followed by Tukey's multiple comparison analysis was performed to determine statistical significance and represented as *$p < 0.05$; ****$p < 0.001$. (I) MitoSOX-based flow cytometric detection of mitochondrial ROS production in shCTRL and shTFEB HeLa cells transfected with siCTRL and siLONP1. Shown is the Mean Fluorescence Intensity (MFI). Shown are mean ± SEM, $n = 3$ biological replicates. One-way ANOVA followed by Tukey's multiple comparison analysis was performed to determine statistical significance and represented as *$p < 0.05$; ****$p < 0.001$. (J) A model showing that S. Typhimurium prevents mTOR-dependent mitochondrial translocation where it interacts with LONP1 to co-regulate complex I assembly and function. Source data are available online for this figure.

matrix-targeting sequence (MTS) (Backes and Herrmann, 2017). MTS is recognized by the receptors TOMM20/TOMM22 and TOMM70 on the mitochondrial surface, however, they are cleaved in the mitochondrial matrix to form mature proteins that fold into their native structure (Backes and Herrmann, 2017). We predicted an MTS sequence within TFEB, which was apparently not necessary for its mitochondrial translocation. However, other mechanisms that involve recognition of a specific consensus motif in the translocase of TOMM20 also enable mitochondrial translocation of proteins (Endo and Kohda, 2002; Ramage et al, 1993). In this study we identified a specific TOMM20 binding consensus motif in TFEB represented as φχχφφ (where φ is a hydrophobic amino acid and χ is any amino acid) (Saitoh et al, 2007). Mutating one amino acid in this binding motif (V to K-30 aa) was sufficient to prevent TFEB mitochondrial translocation. Hence, we termed this motif as mitochondria localization sequence (MLS). The mechanism of translocation was further confirmed by co-immunoprecipitation of both endogenous and ectopically expressed WT-TFEB, with the outer mitochondrial membrane TOMM20, which was lost in the MLS-TFEB mutant, further indicating that the MLS is responsible for its mitochondrial translocation. One of the major hurdles in investigating the mitochondria-centered role of nuclear transcription factors is the dissection of their mitochondrial and nuclear function. Identification of the MLS enabled us to overcome this issue and discern that the nuclear transcriptional function of TFEB was unperturbed even when the MLS was mutated. Interestingly, this motif is conserved in higher mammals, but zebrafish and C. elegans lack this motif, suggesting a derived function arising late in evolution. Consistently, the TFEB orthologue HLH-30 in C. elegans does not translocate to the mitochondria.

The shuttling of transcription factors between the nucleus and mitochondria is regulated by different stress factors (Sepuri et al, 2017). Inhibiting mTORC1 upon starvation or using torin-1 results in the nuclear relocation of dephosphorylated TFEB (Settembre et al, 2012a). We also observed that the interaction of TFEB with mitochondrial proteins is stress-dependent, as it is differentially

modulated upon S. Typhimurium infection and mTOR-inhibition. TFEB subcellular localization is mediated by mTOR-dependent phosphorylation of the serine residues Ser122, Ser142, and Ser211 (Settembre and Ballabio, 2011). Our mass spectrometry analysis revealed that the mitochondrial TFEB is also not phosphorylated. These findings corroborate with our interactomics data, which shows a positive interaction of TFEB with mitochondrial proteins in torin-1-treated cells. In addition, S. Typhimurium prevented TFEB mitochondrial translocation at later time point of infection, which correlates with our previous findings (Ganesan et al, 2017; Tattoli et al, 2012) that it activates mTOR and thereby inhibits autophagy.

Recent reports have indicated that loss of TFEB results in the accumulation of dysfunctional mitochondria (Mansueto et al, 2017). Consistently, TFEB-depleted cells, show altered mitochondrial morphology (more perinuclear mitochondria compared to control cells), and activity. ATP-linked respiration and Maximal respiration were increased in TFEB-depleted cells, which could be restored when the cells were complemented with the WT-TFEB and MTS-TFEB (localized in mitochondria), but could not be rescued with the MLS-TFEB (mutant of TFEB unable to translocate into mitochondria), or with TFEB exclusively localized in cytoplasm or nucleus, which further demonstrated the mitochondria specific function of TFEB in the regulation of OXPHOS. This finding was further supported by mass spectrometry-based quantification of mitochondrial proteins from TFEB-depleted cells, which indicated enhanced expression of complex I subunits of the mitochondrial electron transport chain (ETC). Moreover, complex I activity in TFEB-depleted cells was also increased, suggesting a role for TFEB in regulating complex I function. We specifically found that the NDUFA8 and NDUFB10 subunits were increased in TFEB-depleted cells. Overexpression of TFEB increases complex I-IV activity in PGC1α KO muscle cells and the subunits of complex I (NDUFA9), complex II (SDHA), and complex IV (COX5a) (Mansueto et al, 2017). The discrepancies between studies could be attributed to the cell types studied and the subunits analyzed. Moreover, complementation of TFEB-depleted cells with

the WT-TFEB, restored the elevated levels of NDUFA8 and NDUFB10. However, restoration was not possible when cells were complemented with MLS-TFEB, indicating the regulation of these two complex I subunits by mitochondrial TFEB. Thus, our investigations suggest that TFEB-regulated mitochondrial function is not solely dependent on its transcriptional activity, but also linked to a mitochondria-specific function.

LONP1 is a nuclear-encoded mitochondrial serine protease that regulates proteostasis in mitochondria, mtDNA replication and oxidative phosphorylation (Bota and Davies, 2016; Quiros et al, 2015). LONP1 over-expression results in impaired complex I assembly in melanoma cells, upregulation of NDUFB6,8,10 and 11, and downregulation of NDUFV1, NDUFV2, NDUFS3 and NDUFS7 (Pinti et al, 2016; Quiros et al, 2014). These changes lead to altered respiration and up-regulated glycolysis (Quiros et al, 2014). Consistently, we found that knocking down LONP1 in TFEB-depleted background reduced NDUFA8 and NDUFB10 but restored the oxygen consumption rate and ATP-linked respiration to baseline levels. Interestingly, LONP1 also co-immunoprecipitated with TFEB confirmed by mass spectrometry (Dataset EV1) and immunoblotting which imply that they co-regulate complex I function. These findings suggest that TFEB and LONP1 can regulate either the assembly or turnover of Complex I which will be very interesting to explore in the future.

Increased LONP1 expression reduces the use of NADH equivalents and the respiration through complex I (Santidrian et al, 2013). As a consequence, reactive oxygen species produced by complex I activate p38, JNK, ERK1,2 and Ras-ERK signaling, which favors inflammation (Benhar et al, 2001). Correspondingly, inhibition of LONP1 dampened *S.* Typhimurium-induced inflammatory cytokines. LONP1 inhibitor CDDO has shown promising anti-cancer activity and it is compelling to speculate that the outcome could be influenced by the mitochondrial function of TFEB. Furthermore, enhanced expression of inflammatory cytokines in TFEB-depleted cells was reduced only when complemented with WT-TFEB and MTS-TFEB. Dysfunctions in complex I and ETC not only impact energy metabolism, but are also associated with neurodegenerative diseases (Wallace, 2010), cancer-progression (Urra et al, 2017), and inflammation(Yu et al, 2015). Consistent, with previous reports (Garaude et al, 2016), inhibition of complex I resulted in decreased *S.* Typhimurium-induced IL-6. Therefore, we propose that modulating the mitochondrial localization of TFEB has the potential to mitigate complex I associated diseases. mTOR-dependent mitochondrial translocation of TFEB suggests that the efficacy of mTOR inhibitors used in the clinic as immunosuppressant and in cancer therapy is possibly dependent on the mitochondrial activity of TFEB. In the future, it will be fascinating to extend the physiological relevance of the mitochondria-intrinsic function of TFEB in animal models.

# Methods

## Antibodies, siRNAs, and primers

Antibodies and cell-staining reagents are listed in Table EV1; siRNAs in Table EV2, and primers in Table EV3.

## Cell culture and treatments

Cells were cultured in the following media: HeLa (ATCC) and HEK293T in Dulbecco´s modified Eagle's medium (DMEM) (11960, Life Technologies) supplemented with 10% fetal bovine serum (FBS) (Biowest, S1810-500) and kept at 37 °C in a humidified 5% $CO_2$ incubator. Stable TFEB knockdown was achieved by transfecting TFEB-specific shRNA cloned in pRFP-C-RS plasmid (OriGene RNAi, 0513). The same plasmid containing non-specific shRNA was used as a control (OriGene RNAi, 0415). The cells were transfected using Lipofectamine 3000 (L3000008, Thermo Fisher Scientific) and the clones resistant to puromycin (A11138-03, Life Technologies) were selected. All the constructs were transfected using Lipofectamine 3000. siRNA transfections were performed using 50 nM of siRNA using Lipofectamine RNAiMax (13778030, Thermo Fisher Scientific) according to the manufacturer's recommendations. The indicated drugs were directly added in the culture medium: torin-1 (10 μM) (4247/10, R&D and Tocris), Rotenone (500 nM) (ab143145, Abcam) and CDDO (2.5 μM) (Cay81035, Biomol), Mitotempo (10 μM) (Sigma Aldrich, SML0737).

## Immunocytochemistry

For immunofluorescence, cells previously seeded on coverslips were fixed with 4% PFA (28908, Thermo Scientific) for 15 min, washed three times in PBS, permeabilized in 0.3% Triton-100 followed by incubation with Image-iT® R FX (#I36933, Invitrogen) and blocked with 5% BSA. Cells were then incubated overnight at 4 °C with specific primary antibodies. The following day, after washing with 0.03% Triton in PBS, cells were incubated with the relevant secondary antibody for 1 h at RT protected from light. ProLong® Gold antifade DAPI (P36962, Invitrogen) was used for nuclear counterstaining. The slides were imaged using a 63X objective under a confocal microscope (Leica SP8, Leica Microsystems).

The MitoTracker deep red (M22426, Thermo Fischer Scientific) staining was performed by incubating the cells with 250 nM of Mitotracker deep red in DMEM for 1 h at 37 °C in a humidified 5% $CO_2$ incubator. Coverslips were extensively washed with PBS, prior to mounting with ProLong® Gold antifade DAPI (P36962, Invitrogen).

## mtDNA immunohistochemistry

Cells previously seeded on coverslips were fixed in 4% PFA in PBS for 10 min at RT, and subsequently permeabilized with 0.1% Triton X-100 in PBS for 5 min at RT. After blocking, cells were incubated overnight at 4 °C with the primary antibody against mtDNA (61014, Progen), prepared in 3% BSA in PBS. Cells were then incubated with the secondary goat anti-mouse IgM Alexa594 (Invitrogen, A21044) in 3% BSA in PBS for 2 h at RT. Finally, coverslips were extensively washed with PBS, prior to mounting with ProLong® Gold antifade (P10144, Invitrogen).

## Constructs, cloning, and mutagenesis

For experiments based on the expression of ectopic TFEB, cDNAs were cloned into the p162 vector provided by Dr. Oleg Kurt, to

generate N-terminally FLAG-tagged TFEB proteins. The MLS-TFEB mutant was generated using the Quikchange XL site-directed mutagenesis kit (200521, Agilent). The ΔMTS mutant was generated by cloning a truncated sequence of TFEB in the pcDNA 3.1 FLAG backbone (121416, In-fusion HD Cloning). The TFEB mutants TFEB-ΔNLS (cytosolic TFEB) and S142A-S211A (nuclear TFEB) were provided by Dr. Oleg Kurt.

## Subcellular fractionation assays

Cellular fractionation to prepare cytosolic, mitochondrial, and nuclear extracts was performed using a kit (ab109179, Abcam), according to manufacturer's instructions. The nuclear fraction was finally sonicated. GAPDH, LAMINB and TOMM20 were used as markers for cytosol, nucleus and mitochondria respectively.

## Mitochondria purification using anti-TOMM22 microbeads

HEK293T cells were harvested and mitochondria were purified using anti-TOMM22 labeled magnetic micro-beads according to the manufacturer's instructions (Miltenyi Biotec 130-094-532). 50 μg of purified mitochondria were subjected to Western blot analysis and 10% of total lysate was loaded as positive control.

## Preparation of human monocyte-derived macrophages

After obtaining written consent from healthy donors, blood samples were obtained from two female donors in the age of 20–30. This was approved by the Ethics Commission of the University Hospital of Cologne, Germany (file no: 12-164). Written consent of participants was received from participants before inclusion in the study. All clinical investigations were conducted according to the Declaration of Helsinki principles. CD-14 positive monocytes were isolated from peripheral blood mononuclear cells (PBMCs) and differentiated with M-CSF for 5 days in RPMI/10% FCS for cell culture ex-vivo experiments.

## Protection assay

Purified mitochondria were resuspended in the mitochondria isolation buffer (MIB, 200 mM mannitol, 70 mM sucrose, 10 mM HEPES/KOH pH 7.4, 1 mM EDTA) and incubated with 2.5 μg of trypsin for 15 min on ice. The swelling buffer was prepared in order to cause osmotic shock using 50 mM HEPES/KOH pH 7.2, 0.6 M Sorbitol, and 80 mM KCl. 1% Triton X-100 was eventually added to permeabilize the mitochondrial membranes and allow the trypsin to reach the mitochondrial matrix. Samples were then subjected to SDS-PAGE and immunoblotting. Anti-TOMM20, TIMM23, and SLP2 (or MRPL12 in HEK293T) antibodies were used to analyze trypsin-mediated degradation of proteins located in the outer mitochondrial membrane (OMM), the inner mitochondrial membrane (IMM), and the mitochondrial matrix (MM), respectively.

## In vitro import assay

Purified mitochondria were resuspended in the mitochondria isolation buffer (MIB, 220 mM mannitol, 140 mM Sucrose, 10 mM HEPES/KOH pH 7.4, 1 mM EGTA/KOH).

For in vitro import assay WT and MLS-TFEB cDNA were cloned into the pGEM-4 vector and transcribed in vitro using the TNT Quick coupled translation system (L1170, Promega), by incubating 200 ng of DNA with 7 μL of TNT-T7 reticulocyte lysate mix and 0.5 μL of 35S-labeled methionine (NEG709A500UC, PerkinElmer) for 1 h at 30 °C at 600 rpm. Next, the mitochondrial pellets were resuspended into mitochondrial import buffer (500 mM Sucrose, 10 mM MgAc, 160 mM KAc, 20 mM sodium succinate and 40 mM Hepes-KOH pH 7.4, 0.3 M ATP, and 1 M fresh DTT). Then, mitochondria were incubated with the lysate containing TFEB transcribed in vitro and the membrane potential was blocked by adding 1 μL of 20 mM CCCP at different time points. Mitochondria were also treated with Trypsin (T1426, Sigma Aldrich) (25 μg/mL for 10 min) followed by trypsin inhibitor soya bean (Merck, 10109886001) (250 μg/mL for 15 min). 1% Triton X-100 was eventually added to permeabilize the mitochondrial membranes and allow the trypsin to reach the mitochondrial matrix. The mitochondrial pellet was lysed in denaturing conditions and proteins were detected by autoradiography.

## Seahorse XFe96 metabolic flux analysis

Cellular respiration was measured using the Seahorse XFe96 analyzer (Agilent) following manufacturer's recommended protocols. Briefly, HeLa cells were seeded in a 6 well-plate and transfected with FLAG, WT-TFEB FLAG, MLS-TFEB, NLS-TFEB and NL-TFEB FLAG plasmids, or with siCTRL and siLONP1. After 24 h, cells were seeded in a 96 well-plate and the next day the plates were incubated in 180 μL of assay media at 37 °C in a $CO_2$ free incubator for 1 h. The plates were assayed on the XFe96 analyzer to calculate the oxygen consumption rate (OCR) over time after sequential addition of different drugs. Their final concentrations in the assay were: Oligomycin (6 μM), FCCP (0.5 μM), Rotenone (0.5 μM), and Antimycin A (0.5 μM).

HEK293T cells, were seeded in a 6 well-plate and transfected with siRNA CTRL and siRNA against TFEB. After 24 h, cells were seeded in a Poly-D-Lysine pre-coated 96 well-plate. Next day, the plates were incubated in 180 μL of assay media at 37 °C in a $CO_2$ free incubator for 1 h. The plates were assayed on the XFe96 analyzer to calculate the oxygen consumption rate (OCR) over time after sequential addition of different drugs. Their final concentrations in the assay were: Oligomycin (6 μM), FCCP (0.3 μM), Rotenone (0.5 μM), and Antimycin A (0.5 μM). After the run, the assay media was removed, and the cells were lysed for analysis. The data were normalized with protein estimated by Bradford assay.

## Western blotting

Western blotting was performed on proteins extracted using RIPA buffer containing complete protease and phosphatase inhibitor cocktail. Samples were incubated on ice for 20 min and centrifuged at 13,200 rpm for 10 min. BCA was done to quantify the amount of proteins in the lysates. Required samples were mixed 1:1 with 2X sample loading buffer, boiled at 95 °C, and resolved by SDS-PAGE. Proteins were then transferred onto a PVDF membrane blocked with 5% milk or BSA and probed with the respective primary antibodies, followed by treatment with an appropriate secondary antibody conjugated to horseradish peroxidase. The blots were developed using an enhanced chemiluminescence substrate (GE

Health sciences) and bands were identified by exposing the membrane on to an X-ray film or imaged using the ChemiDoc Imaging System (Biorad). Densitometric analysis of immunoblots was performed using NIH ImageJ.

## *Salmonella* Typhimurium infection

HeLa cells were seeded in tissue culture plates and infected with *S.* Typhimurium (SL1344) (MOI 100). After 30 min, extracellular bacteria were removed and cells were incubated for 2 h in medium containing 50 μg/ml Gentamicin (15750037, Life Technologies). Then, the cells were washed and subsequently cultured in medium containing lower concentration of Gentamicin (10 μg/ml). Cells were collected for analysis at the desired time points. All experiments with S. Typhimurium infection were handled in a Biosafety Level 2 laboratory and the protocol was approved by University of South Australia's Institutional Biosafety Committee (Reference No: IBC-B-032).

## mtROS detection

mtROS was determined with MitoSOX (M36008, Thermo Fisher Scientific) according to the manufacturer's instructions. At the indicated time points, HeLa cells ($0.5 \times 10^6$ cells/well) were resuspended in 1% (wt/vol) PFA in PBS. For each experiment, 50,000 cells were acquired using a flow cytometer (FACSCanto; Becton Dickinson) and analyzed using FLOWJO software (Tree Star).

## RNA extraction and quantitative PCR

Total RNA was extracted from cells in QIAzol reagent (79306, Qiagen), using RNeasy kit (74106, Qiagen). RNA was reverse transcribed using the iScript cDNA synthesis kit (1708891, Biorad). The resulting cDNA was quantified by real-time qPCR using the LightCycler 480 SYBR Green Master Mix (4367659, Thermo Fisher Scientific). The relative mRNA levels were calculated using the ΔΔCt method and normalized to Hypoxanthine-Guanine-Phosphoribosyl-Transferase (HPRT) or to Ribosomal Protein L13, (RPL13A) mRNA levels.

## Co-immunoprecipitations

For endogenous immunoprecipitations, $10^7$ cells were lysed in cell lysis buffer (98035S, Cell Signaling Technology) containing protease inhibitors (4693159001, Roche) and incubated for 15 min on ice. Immunoprecipitation was performed by incubating 1 mg of the whole cell lysates (input) with 2 μg of antibody overnight at 4 °C. Isotype specific rabbit IgG (2729, Cell Signaling Technology) was used as negative control. The following day, protein A sepharose beads (101041, Thermo Fischer Scientific) were added and incubated for 2 h at 4 °C. The immunocomplexes were then washed 3 times with lysis buffer, resuspended in Laemli buffer, and samples were boiled at 95 °C for 5 min and processed for immunoblotting.

For the ectopic immunoprecipitations, $10^7$ cells overexpressing TFEB-FLAG and LONP1 were lysed in cell lysis buffer (98035S, Cell Signaling Technology) containing protease inhibitors (4693159001, Roche) and incubated for 15 min on ice. Immunoprecipitation was performed by incubating 1 mg of the whole cell lysates (input) with M2 mouse anti-FLAG (F3165, Sigma Aldrich) and 50 μL of microMACS Protein G magnetic beads (Miltenyi, 130-

042-701). Samples were washed with 150 μL 50 mM Tris-HCl pH 7.4, 1% NP-40 and then eluted into Laemli buffer, and samples boiled at 95 °C for 5 min and processed for immunoblotting.

## Protein purification and interaction

TFEB-FLAG and LONP1-HA were overexpressed in HEK293T cells and were harvested after 24 h. The pellets were lysed into Lysis buffer (20 mM Tris-HCl pH 7.5, 150 mM NaCl, 1 mM EDTA, 1 mM EGTA, 1% Triton, 2.5 mM sodium pyrophosphate, 1 mM b-glycerophosphate, 1 mM Na3 VO4 and protease inhibitors (4693159001, Roche), incubated for 15 min on ice and centrifuged at $17,000 \times g$ for 10 min and cleared lysates were collected. CTRL vector or TFEB-FLAG lysates were incubated with M2 anti-FLAG agarose beads (A22220, Sigma Aldrich) for 3 min at 4 °C on wheel. The beads were then washed 4 times with Buffer A: 50 mM Tris-HCl pH 7.4, 150 mM NaCl, 0.05% Tx100, 10% glycerol, 1 mM EDTA, 2.5 mM sodium pyrophosphate, 1 mM b-glycerophosphate and 1 mM $Na_3VO_4$.

LONP1 lysate was incubated with anti-HA antibody and protein G sepharose beads (20398, Thermo Fischer Scientific) for 30 min at 4 °C on wheel. Beads were washed 4 times with Buffer A and samples were eluted with 1mg/mL of HA-peptide in buffer Aby incubating 30 min at 4 °C on wheel. The CTRL and TFEB-FLAG beads were added to the purified LONP1 and the samples were eluted in Laemli buffer.

## Staining in *C. elegans*

The strain used for the mitochondrial staining is *hlh-30(syb809) IV* which has the endogenous *hlh-30* locus tagged with mNeonGreen by CRISPR/Cas9. Synchronized eggs were incubated on NGM agar plates with *E. coli* strain OP50 containing 500 nM MitoTracker Deep Red FM (Invitrogen M22426) until day 1 adults. The worms were mounted on 5% agarose pads and anesthetized with 20mM levamisole. Pictures were taken with confocal laser scanning microscopy platform Leica TCS SP8.

## Blue native

For blue native electrophoresis, mitochondria were solubilized in lysis buffer (50 mM NaCl, 10% glycerol, 5 mM 6-aminohexanoic acid, 50 mM imidazole/HCl pH 7.0, 50 mM KP$_i$ buffer pH 7.4) containing 1% (w/v) digitonin for 30 min on ice. Insoluble material was removed by centrifugation and subjected to blue native electrophoresis.

## In gel enzymatic activity staining

Purified mitochondria were subjected to blue native electrophoresis and the gels were rinsed briefly in MilliQ water and then equilibrated in reaction buffer (0.1 M Tris-HCl pH 7.4) without substrates for 10 min. The gels were then incubated at room temperature in fresh buffer plus reagents (0.2 mM NADH and 1 mg/mL NBT) for varying lengths of time until sufficient staining.

## Electron microscopy

HeLa cells ($2 \times 10^6$ cells/well) were fixed by immersion in 2% glutaraldehyde, 2.5% Sucrose, 3 mM CaCl$_2$ and 100 mM HEPES pH

7.4 for 30 min at room temperature and 30 min at 4 °C. After the fixation the cells were washed three times in 0.1 M Sodium Cacodylate buffer. After treatment with Osmium for 1 h on ice, the sections were dehydrated with ethanol and embedded in epoxy resin. The cells were then cut 70 nm thick on a Ultracut from Leica. They were contrasted with 1.5% uranyl acetate for 15 min at 37 °C in the dark, then washed with water 5 times and washed with leadcitrate for 4 min. Finally, the grids were washed 5 times in water and dried on a filter paper. The cells were cut 70 nm thick on an Ultracut from Leica; they were contrasted with 1.5% uranyl acetate for 15 min at 37 °C in the dark, washed with water 5 times and then with leadcitrate 4 min. Finally, the grids were washed 5 times in water and then dried on a filter paper. Samples were examined with the electron microscope JEOL JEM 2100 plus.

## Electron microscopy analysis

To test for preferential labeling of compartments, observed and expected gold particle distributions are compared by χ2 analysis. Efficient estimators of gold labeling intensity (relative labeling index (RLI) were calculated to analyze volume-occupying compartments (cytosol, mitochondria and nucleus). Compartment size is estimated by counting chance events after randomly superimposing test lattices of crosses. RLI = 1 when there is random labeling and RLI >1 when there is preferential labeling.

## Confocal image analysis

Colocalization was analyzed using Colocalization Analysis plugins (ImageJ software). First, the Colocalization Test plugin with Fay randomization method was performed to calculate Pearson's correlation coefficient for the two channels in each selected ROI ($25 \times 25$ pixels). This value was compared with what would be expected for random overlap. The observed correlation was considered significant if it was greater than 95% of the correlations between channel 1 and a number of randomized channel 2 images. All ROIs with Pearson's coefficient $p$ value of ≥0.95 were further analyzed by the Colocalization Threshold plugin to calculate thresholded Mander's coefficients [tM1 colocalization value for channel 1 (red); tM2 colocalization value for channel 2 (green or blue)] and to generate scatterplots with linear regression line and thresholds.

## ELISA

Cell culture supernatants from HeLa cells ($5 \times 10^7$ cells/well) were collected 24 h after infection and were kept at −80 °C until assayed for human IL-6 (DY206-05, R&D Systems) and human IL-8 (DY208-05, R&D) according to the manufacturer's instructions.

## Interactome analysis

TFEB-FLAG IP for immunoblot and mass spectrometry was performed in triplicates for each condition using Anti-FLAG M2 Magnetic Beads, according to the manufacturer's instructions. Washed beads with enriched proteins were denatured in 6 M urea/2 M thio-urea in 10 mM HEPES buffer. Proteins were reduced by dithiothreitol (DTT) at a final concentration of 10 mM for 30 min at 25 °C. Then, proteins were alkylated using Iodacetamide (IAA) at a final concentration of 55 mM in the dark. Proteins were

separated by molecular weight on an SDS-PAGE. Gel lanes were cut into 5 equal sized pieces and further cut into 1mm² cubes. Gel pieces were washed several times and proteins were digested in-gel using a digestion solution (12 ng/µL Trypsin, 2 ng/µL LysC) was added to cover gel pieces. Generated peptides were extracted by an increasing acetonitrile content. Peptides were concentrated in a SpeedVac to 50 µL. 100 µL 0.1% formic acid was added and samples were desalted using the STAGE tip technique.

## Mitochondrial proteome

Mitochondria were isolated in the MIB, and the pellet was resuspended in the following buffer: 50 mM TrisHCl pH 8.0, 2% SDS, 2% CHAPS and 1% NP-40. 500 µg of protein was precipitated by adding 4 volumes of acetone and incubation at −20 °C for at least 15 h. The pellet was washed with 90% acetone and resuspended in 6 M Urea/2M Thio-Urea in 10 mM HEPES. Proteins were reduced using 10 mM Tris (2-carboxyethyl) phosphine (TCEP), alkylated with 20 mM chloroacetamide (CAA) and digested using 1 µL of 0.5 µg/µL of Lys-C at a 1:100 enzyme to substrate ratio to the mixture for 2 h. Then 50 mM Ammonium bicarbonate was used to dilute the Urea concentration to 2 M supplemented with 1 µL of 0.5 µg/µL of Trypsin. Overnight digestion was stopped by Trifluoroacetic acid (TFA) acidification (1% final concentration).

## TFEB phosphorylation detection

Endogenous TFEB was pulled down in both cell lysates and purified mitochondria. Next, samples were loaded on SDS-PAGE and proteins were separated according to their molecular mass. Gels were stained with InstantBlue for 30 min and gel bands were cut of the matching molecular weight. Proteins were digested in-gel and peptides were desalted prior LC-MS/MS analysis.

## Liquid chromatography and mass spectrometry measurements

Peptides were eluted from home-made StageTips using 60% acetonitrile and 0.1% formic acid and evaporated to complete dryness using a SpeedVac (Eppendorf). Peptides were reconstituted in 2% formic acid and 2% acetonitrile. The LC-MS/MS equipment consisted out of an Easy nLC 1000 (Thermo Fisher Scientific) coupled to a QExactive Plus mass spectrometer via a nano-spray ionization source. Peptides were separated according to their hydrophobicity using in-house made columns (ReproShell C18, Agilent, 2.7 µm). Column temperature was controlled by an in-house made column oven. Peptides were ionized using a Spray Voltage of 2.4 kV and the mass spectrometer operated in a data-dependent mode (Top10). The MS1 spectra were acquired at a resolution of 70,000 (at 200 $m/z$) and a maximum injection time of 20 ms. The mass range was 350–1750 $m/z$.

The MS2 settings varied depending on the experiment. For whole proteome and interactome (in-solution) analysis, a resolution of 17,500 at 200 $m/z$ and a maximum injection time of 60 ms was used. For detection of phosphorylation sites from pulled down endogenous TFEB, a higher spectra quality was required and a resolution of 30,000 at 200 $m/z$ accompanied with a maximum injection time of 120 ms was utilized.

## Raw data processing and analysis

Acquired raw data were analyzed using MaxQuant (Cox and Mann, 2008) and the implemented search engine Andromeda (Cox et al, 2011). MS/MS spectra were correlated to the human Uniprot reference proteome using default settings for mass tolerances. The 'match-between-runs' and MaxLFQ algorithms were enabled using default settings. For detection of phosphorylation sites, a minimum score of 0 was used but MS/MS spectra of detected TFEB sites were inspected manually.

Contaminants, 'only identified by site' and 'reverse' data base hits were removed in Perseus. LFQ intensities or site intensities were $\log_2$ transformed. Significantly different regulated/enriched proteins were determined by a two-sided $t$-test using a permutation-based FDR estimation (s0 = 0.1, number of permutations: 500) for whole proteome and interaction studies. Gene Ontology annotations were added from the Uniprot database. Data were analyzed and visualized in Instant Clue (Nolte et al, 2018). Data were uploaded to PRIDE repository and can be found under the following identifier: PXD013758.

## Statistical analysis

All the statistical analysis was performed using GraphPad Prism software. To determine statistical significance One way ANOVA with Tukey Posthoc test was conducted where three or more groups are compared and two-tailed Student's t-test was conducted when only two groups were compared. All data are represented as mean ± SEM as indicated. For all tests, $p$-values < 0.05 was considered statistically significant (*$p < 0.05$; **$p < 0.01$; ***$p < 0.005$; ****$p < 0.001$).

# Data availability

The mass spectrometry proteomics data have been deposited to the ProteomeXchange Consortium via the PRIDE partner repository with the dataset identifier PXD013758. The data can be accessed using the following details. Project Name: TFEB Immunoprecipitation and Proteome. Reviewer account details: Username: reviewer97397@ebi.ac.uk, Password: 1xuY7MRD.

# Peer review information

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

## Acknowledgements

We thank the Robinson laboratory members for helpful discussions; Dr. Christian Jüngst, Dr. Astrid Schauss and Ms Beatrix Martiny from the Cluster of Excellence in Cellular Stress Responses in Aging-associated Diseases (CECAD) Imaging Facility, University of Cologne, for assistance with electron microscopy and confocal microscopy; Prof. Thomas Langer (MPI-AGE) for the helpful discussions on mitochondrial experiments; Dr. Oleg Krut and Mr. Paul Moretti for their help in constructing plasmids. Zhara Hejazi for technical support; Dr. Tom MacVicar for assistance with the import assay; Pablo Rivera for the assistance with Sea horse and antibody supply. Dr. Rebecca George Tharyan for scientific discussion. This work was funded by Deutsche Forschungsgemeinschaft through CECAD and SFB670, Centre for Cancer Biology and University of South Australia (to N.R.), the Max Planck Institute for the Biology of Ageing (to A.A.); Cologne Graduate School of Ageing Research (to C.C. and R.L.); DZIF (German Center of Infection Research) (to J.F.).

## Author contributions

**Chiara Calabrese**: Conceptualization; Formal analysis; Investigation; Methodology; Writing—original draft; Writing—review and editing. **Hendrik Nolte**: Data curation; Formal analysis; Investigation. **Melissa R Pitman**: Formal analysis; Investigation; Methodology. **Raja Ganesan**: Formal analysis; Investigation. **Philipp Lampe**: Formal analysis; Investigation. **Raymond Laboy**:

Formal analysis; Investigation. **Roberto Ripa**: Formal analysis; Investigation. **Julia Fischer**: Investigation. **Ruhi Polara**: Formal analysis; Investigation. **Sameer Kumar Panda**: Formal analysis; Investigation. **Sandhya Chipurupalli**: Investigation. **Saray Gutierrez**: Conceptualization; Formal analysis. **Daniel Thomas**: Resources; Writing—review and editing. **Stuart M Pitson**: Resources; Supervision; Writing—review and editing. **Adam Antebi**: Conceptualization; Resources; Writing—original draft; Project administration; Writing—review and editing. **Nirmal Robinson**: Conceptualization; Resources; Formal analysis; Supervision; Funding acquisition; Investigation; Methodology; Writing—original draft; Project administration; Writing—review and editing.

## Disclosure and competing interests statement

The authors declare no competing interests.

