## [Peer Review File · EMBO Reports]

Mitochondrial translocation of TFEB regulates complex I and inflammation

Chiara Calabrese, Hendrik Nolte, Melissa Pitman, Raja Ganesan, Philipp Lampe, Raymond Laboy, Roberto Ripa, Julia Fischer, Ruhi Polara, Sameer Panda, Sandhya Chipurupalli, Saray Gutierrez, Daniel Thomas, Stuart Pitson, Adam Antebi, and Nirmal Robinson

DOI: [10.15252/embr.202357116](https://doi.org/10.15252/embr.202357116)

Corresponding author(s): Nirmal Robinson (nirmal.robinson@unisa.edu.au), Adam Antebi (AAntebi@age.mpg.de)

Review Timeline:

Submission Date:	3rd Mar 23
Editorial Decision:	23rd Mar 23
Appeal Received:	3rd Apr 23
Editorial Decision:	3rd Apr 23
Revision Received:	16th Jul 23
Editorial Decision:	5th Oct 23
Revision Received:	6th Dec 23
Accepted:	22nd Dec 23

Editor: Ioannis Papaioannou / Martina Rembold

Transaction Report:

Dear Dr. Robinson,

Thank you for transferring your research manuscript for consideration by EMBO reports. Your manuscript has now been seen by two experts in the field, and we have received their detailed reports (included below).

As you will see, both referees acknowledge that the observed mitochondrial localization of TFEB is interesting. However, both referees also identify important limitations and raise several significant concerns regarding the need for further insight into TFEB-mediated non-transcriptional regulation of mitochondrial respiration, for addressing contradictory results with previous literature, for validating results *in vivo*, for excluding the possibility of mitophagy impairment upon TFEB depletion, for confirming results in the absence of TFEB (full knockout), and for investigating the localization and function of other MITF/TFE family transcription factors in mitochondria. Both referees also identify a number of limitations regarding missing controls from some experiments, the low quality of some data, and the analysis and presentation of the results. Due to the many criticisms and technical problems of the current version, and the amount of experimental work likely to be required to address them, I am afraid that we do not feel it would be productive to call for a revised version of your manuscript.

Given the potential interest of the findings, we would, however, have no objection to consider a new manuscript on the same topic if at some time in the near future you obtained data that would considerably strengthen the study and address the referees' concerns in full. If you were to send a new manuscript that would address the concerns put forward by the referees, it would be treated as a new submission rather than a revision, and it would be reviewed afresh, also with respect to any novel literature on the topic at the time of submission.

I am sorry to disappoint you on this occasion, but I nevertheless hope that you will find the referees' comments and suggestions helpful in your work. I would like to thank you once again for your interest in our journal and for the opportunity to consider your manuscript.

Yours sincerely

Ioannis Papaioannou, PhD
Editor
EMBO reports

Referee #1:

TFEB is a regulator of lysosome biogenesis, autophagy and mitochondrial metabolism that mainly function on transcriptional level. Calabrese and colleagues report that TFEB also localizes to mitochondria. They used an array of independent assay to determine the localization. Using a TFEB mutant that are impaired in mitochondrial localization, the authors found that loss of the mitochondrial pool of TFEB affects mitochondrial morphology and respiration. Here, TFEB antagonistically functions with the protease LON in the regulation of complex I and inflammatory response. Upon infection with *Salmonella Typhimurium*, the lack of TFEB increases the expression of pro-inflammatory cytokines.

The identification of the mitochondrial pool of TFEB is very interesting and opens new directions for further studies on the mechanisms how TFEB operates. The authors provide convincing evidence for the mitochondrial localization. Overall, the study is well done, but requires a number of control experiments and extensive proof reading.

1. Does TFEB also affects transcription of mitochondrial genes?
2. The authors should provide Volcano plots for Figure 1 to provide an overview of the mitochondrial proteins interacting with TFEB.
3. The authors provide a number of interesting data with cells expressing a TFEB variant that lacks the putative mitochondrial targeting signal. To define which functions are not related to the mitochondrial pool of TFEB, the authors should express a TFEB variant that contains a strong mitochondrial targeting signal.
4. The shown Seahorse experiments are of poor quality. The color code of the two subfigures should be adjusted. The in-gel activity stain does not allow quantitative assessment. The author should provide additional data to show the activity of the respiratory chain complexes.
5. The blue native gel in Figure 5 is of poor quality and should be repeated.

6. The authors should show negative controls in the co-immunoprecipitations in Figure 6.
7. The submitochondrial localization of TFEB is not entire clear. Why does TFEB interacts with TOMM20 (Figure 3I) when it localizes to the matrix? Furthermore, the localization of TFEB close to mtDNA is not convincing. These points should to be clarified.
8. The authors should add a quantification to the localization of TFEB in Figure 3B.
9. Figure 3C: Import of wild-type TFEB should be included as control.
10. 3H. The authors state: "Ser467 was the only phosphorylated residue in the mitochondrial pool.... The phosphorylation of TFEB in these Ser residues was lost upon Torin treatment." The data presented here show the opposite effect.
11. The manuscript has to be checked in detail. E.g. first sentence of abstract which is mixed up, . mTOR is "mammalian" not "mechanistic" target of rapamycin, the terms "expression" and "steady state level" are often mixed.

 Referee #2:

In this manuscript, the authors discuss the role of TFEB, a master regulator of autophagy, lysosome biogenesis, and mitochondrial metabolism, in regulating the electron transport chain complex I in mitochondria. The study shows that TFEB has a non-transcriptional role in down-modulating inflammation through its interactions with several mitochondrial proteins. The localization of TFEB in the mitochondrial matrix was observed, and it was found that TFEB and protease LONP1 co-regulate complex I, reactive oxygen species, and the inflammatory response. Lack of TFEB specifically in the mitochondria during infection leads to the worsening of pro-inflammatory cytokine expression and contributes to innate immune pathogenesis. Although the observed mitochondrial localization of some TFEB proteins is interesting, the contradicting results observed in this study with previous literature regarding the role of TFEB in mitochondrial function need to be addressed and validated in in vivo models. The mechanism of TFEB's non-transcriptional regulation of mitochondrial respiration should also be investigated. Additionally, the authors need to exclude the possibility of mitophagy impairment upon TFEB depletion, and critical experiments should be replicated in full KO for TFEB. Finally, the localization and role of other MITF/TFE family transcription factors, such as MITF and TFE3, within the mitochondria need to be explored. Overall, the manuscript needs to address these concerns and provide more evidence to support its claims.

Other important comments and concerns:

Regarding figures in general the authors should plot the graphs showing individual replicates, not the bars. Regarding the legends, authors often do not write which cell line they use for the experiments.

Figure 1 D: I think IF of endogenous TFEB is very weak like a simple background

Figure 1E: the scale should be reduced to 0,4. There are no statistics.

Figure 1F, H: Asking triplicates with densitometry and statistics

Legend to Figure 1 F-H: cell line is not indicated

Figure 2B: WB in triplicate with densitometry and statistics

Figure 2C: there are no statistics

Figure 2D: low-resolution image

Figure 2G: graph in dots of replicates, instead of bars, to add statistics

Figure 3A: leave only one graphical representation, instead of two

Figure 3B: WB in triplicate with densitometry and statistics

Figure 3E: it might be 1-way ANOVA (or 1 sample t-test)

Figure 3F: they can add also a confocal image of TFEB in mitochondria under Totin1 treatment in comparison to untreated cells

Figure 4B: Not clear the differences in morphology, no quantification, and stats

Figure 4C: to add statistics

Figure 4D: dot-plot instead of bars on the graph. Might be 1 way ANOVA (or 1 sample t-test)

Figure 6B: To change dots instead of bars on the graph. Might be 1 way ANOVA (or 1 sample t-test)

Figure 6D: To change dots instead of bars on the graph.

Figure 6E: to add statistics to OCR

Figure 6F: To change dots instead of bars on the graph. Might be 1 way ANOVA (or 1 sample t-test)

Legend to Figure 6C, F, I: cell line is not written

Figure 7 C-E: To change dots instead of bars on the graph.

Figure 7E-I: Might be 1-way ANOVA (or 1 sample t-test)

Legend to figure 7F-I: cell line is not indicated.

- Abbreviation to TFEB, ROS, mTOR, but not too many other proteins, like Optn, STAT3, etc.

- In the Abstract the 1st sentence is strange: "that works, and immunity, primarily through" and immunity is placed not in logical way.

In the introduction, line 54, they should cite the new link about TFEB and Rags interaction obtained from crystallography data from the recent Nature paper.

Introduction Line 75: PMID is inserted by mistake

In the Results Line 149: mistake in the name of cell line: HEK293T instead of HEK293

** As a service to authors, EMBO Press provides authors with the ability to transfer a manuscript that one journal cannot offer to publish to another journal, without the author having to upload the manuscript data again. To transfer your manuscript to another EMBO Press journal using this service, please click on Link Not Available

Dear Ioannis,

I once again thank you for getting our manuscript reviewed and truly appreciate the comments from the reviewers.

As per our discussion we had through zoom, we will be able to address most of the comments raised by the reviewers in a reasonable amount of time (within 2-3months). However, as discussed we will not be able to study the physiological significance of TFEB's function in mitochondria using TFEB KO mouse models for the following reasons.

1. TFEB KO mice are embryonically lethal
2. Inducible KO animals are not widely available
3. Ideally, we would like to generate at TFEB MLS mutant knock-in animals to dissect the mitochondria specific function of TFEB. However, there are so many uncertainties associated with this.

Therefore, I kindly request you to consider my appeal and give us an opportunity to revise this manuscript. Please consider it for a revision and not as a new submission.

I look forward to hearing from you.

Kind regards,

Nirmal

Dear Nirmal,

Thank you for your message asking us to reconsider our decision on your manuscript following our discussion. Taking into consideration the comments of the referees and your expressed willingness to address most of their concerns, we would like to invite you to revise your manuscript with the understanding that the referee concerns (as detailed in their reports) must be addressed and their suggestions taken on board.

In particular, although we agree with referee #2 that the analysis of the physiological significance of TFEB function in mitochondria using a TFEB knockout mouse model would be desirable to strongly substantiate the conclusions of the study, it will not be required for further consideration of your manuscript by EMBO reports. You are, however, expected to address all remaining concerns and use the suggestions of both referees to strengthen and improve your study and your manuscript.

Please address all referee concerns in a complete point-by-point response. Acceptance of the manuscript will depend on a positive outcome of a second round of review. It is EMBO reports policy to allow a single round of revision only and acceptance or rejection of the manuscript will therefore depend on the completeness of your responses included in the next, final version of the manuscript. If you have any questions or comments, we can also discuss the revisions in a video chat, if you like.

We realize that it is difficult to revise to a specific deadline. In the interest of protecting the conceptual advance provided by the work, we usually recommend a revision within 3 months (July 2nd). Please discuss with me the revision progress ahead of this time if you require more time to complete the revisions.

IMPORTANT NOTE:

We perform an initial quality control of all revised manuscripts before re-review. Your manuscript will FAIL this control and the handling will be DELAYED if the following APPLIES:

- 1) If a data availability section providing access to data deposited in public databases is missing (please see below for more information).
- 2) If your manuscript contains statistics and error bars based on $n=2$. Please use scatter plots in these cases. No statistics should be calculated if $n=2$.

- 1) A .docx formatted version of the manuscript text (including legends for main figures, EV figures and tables). Please make sure that the changes are highlighted to be clearly visible.
- 2) Individual production quality figure files as .eps, .tif, .jpg (one file per figure). Please download our Figure Preparation Guidelines (figure preparation pdf) from our Author Guidelines pages <https://www.embopress.org/page/journal/14693178/authorguide> for more info on how to prepare your figures.
- 3) A .docx formatted letter INCLUDING the reviewers' reports and your detailed point-by-point responses to their comments. As part of the EMBO Press transparent editorial process, the point-by-point response is part of the Review Process File (RPF), which will be published alongside your paper unless you opt out of this (please see below for further information).
- 4) A complete author checklist, which you can download from our author guidelines (<<https://www.embopress.org/page/journal/14693178/authorguide>>). Please insert information in the checklist that is also reflected in the manuscript. The completed author checklist will also be part of the RPF (please see below for more information).
- 5) Please note that all corresponding authors are required to supply an ORCID ID for their name upon submission of a revised manuscript (<<https://orcid.org/>>). Please find instructions on how to link your ORCID ID to your account in our manuscript tracking system in our Author guidelines (<<https://www.embopress.org/page/journal/14693178/authorguide#authorshipguidelines>>)
- 6) We replaced Supplementary Information with Expanded View (EV) Figures and Tables that are collapsible/expandable online. A maximum of 5 EV Figures can be typeset. EV Figures should be cited as "Figure EV1, Figure EV2" etc... in the text and their

respective legends should be included in the main text after the legends of regular figures.

7) Before submitting your revision, primary datasets (and computer code, where appropriate) produced in this study need to be deposited in appropriate public databases (see <<https://www.embopress.org/page/journal/14693178/authorguide#dataavailability>>). Specifically, we would kindly ask you to provide public access to the Liquid Chromatography - Mass Spectrometry data reported in your manuscript.

The accession numbers and database should be listed in a formal "Data availability " section (placed after Materials & Methods) that follows the model below (see also <<https://www.embopress.org/page/journal/14693178/authorguide#dataavailability>>):

Data availability

- RNA-seq data: Gene Expression Omnibus GSE46843 (<https://www.ncbi.nlm.nih.gov/geo/query/acc.cgi?acc=GSE46843>)
- [data type]: [name of the resource] [accession number/identifier/doi] ([URL or identifiers.org/DATABASE:ACCESSION])

*** Note: all links should resolve to a page where the data can be accessed. ***

*** Note: the Data Availability Section is restricted to new primary data that are part of this study. ***

8) We request authors to consider both actual and perceived competing interests. Please review the new policy (<<https://www.embopress.org/competing-interests>>) and update your competing interests statement if necessary. Please name this section 'Disclosure and competing interests statement' and place it after the Acknowledgements section.

9) Figure legends and data quantification:

- the name of the statistical test used to generate error bars and P values,
- the number (n) of independent experiments (please specify technical or biological replicates) underlying each data point,
- the nature of the bars and error bars (s.d., s.e.m.)
- If the data are obtained from n {less than or equal to} 2, use scatter plots showing the individual data points.

Discussion of statistical methodology can be reported in the Materials and Methods section, but figure legends should contain a basic description of n, P and the test applied.

10) We now request publication of original source data with the aim of making primary data more accessible and transparent to the reader. Our source data coordinator will contact you to discuss which figure panels we would need source data for and will also provide you with helpful tips on how to upload and organize the files.

11) Our journal encourages inclusion of *data citations in the reference list* to directly cite datasets that were re-used and obtained from public databases. Data citations in the article text are distinct from normal bibliographical citations and should directly link to the database records from which the data can be accessed. In the main text, data citations are formatted as follows: "Data ref: Smith et al, 2001" or "Data ref: NCBI Sequence Read Archive PRJNA342805, 2017". In the Reference list, data citations must be labeled with "[DATASET]". A data reference must provide the database name, accession number/identifiers and a resolvable link to the landing page from which the data can be accessed at the end of the reference. Further instructions are available at <<https://www.embopress.org/page/journal/14693178/authorguide#referencesformat>>.

12) Please also note our reference format:

<<http://www.embopress.org/page/journal/14693178/authorguide#referencesformat>>.

13) We now use CRediT to specify the contributions of each author in the journal submission system. CRediT replaces the author contribution section, which should be removed from the manuscript. Please use the free text box to provide more detailed descriptions. See also guide to authors:

<<https://www.embopress.org/page/journal/14693178/authorguide#authorshipguidelines>>.

14) As part of the EMBO publications' Transparent Editorial Process, EMBO reports publishes online a Review Process File to accompany accepted manuscripts. This File will be published in conjunction with your paper and will include the referee reports, your point-by-point response and all pertinent correspondence relating to the manuscript.

You can opt out of this by letting the editorial office know (emboreports@embo.org). If you do opt out, the Review Process File link will point to the following statement: "No Review Process File is available with this article, as the authors have chosen not to make the review process public in this case."

I look forward to seeing a revised version of your manuscript when it is ready. Please let me know if you have any questions or comments regarding the revision.

Best regards,

Ioannis

Ioannis Papaioannou, PhD
Editor
EMBO reports

Referee #1:

TFEB is a regulator of lysosome biogenesis, autophagy and mitochondrial metabolism that mainly function on transcriptional level. Calabrese and colleagues report that TFEB also localizes to mitochondria. They used an array of independent assay to determine the localization. Using a TFEB mutant that are impaired in mitochondrial localization, the authors found that loss of the mitochondrial pool of TFEB affects mitochondrial morphology and respiration. Here, TFEB antagonistically functions with the protease LON in the regulation of complex I and inflammatory response. Upon infection with *Salmonella Typhimurium*, the lack of TFEB increases the expression of pro-inflammatory cytokines.

The identification of the mitochondrial pool of TFEB is very interesting and opens new directions for further studies on the mechanisms how TFEB operates. The authors provide convincing evidence for the mitochondrial localization. Overall, the study is well done, but requires a number of control experiments and extensive proof reading.

We are pleased to note that the referee found our work interesting and found our data on TFEB mitochondrial localization convincing. We hope to have thoroughly addressed the points raised by the referee to enhance the quality of the paper.

1. Does TFEB also affects transcription of mitochondrial genes?

We analysed the mRNA levels of mitochondrial genes, and we did not observe any significant differences between the ctrl and TFEB-depleted cells. This data is presented in Fig. S6E. We also analysed the mRNA expression of LONP1, NDUFA8, NDUFA10, NDUFS1, NDUFS5 and the mitochondria encoded mtND6 upon the loss of TFEB or complemented with mutant versions of TFEB and did not observe any significant difference (Fig S4I).

2. The authors should provide Volcano plots for Figure 1 to provide an overview of the mitochondrial proteins interacting with TFEB.

We have provided the Volcano plot in Figure S1A.

3. The authors provide a number of interesting data with cells expressing a TFEB variant that lacks the putative mitochondrial targeting signal. To define which functions are not related to the mitochondrial pool of TFEB, the authors should express a TFEB variant that contains a strong mitochondrial targeting signal.

We thank the reviewer for this suggestion. In the submitted manuscript we had dissected the role of mitochondria localised TFEB using 2 mutant forms of TFEB, one that localised only in cytoplasm and the other localised in nucleus. We have now performed extensive experiments to delineate the mitochondrial function of TFEB. We succeeded in tagging the Mitochondrial Targeting Sequence (MTS) of the mitochondrial protein SOD2 with TFEB (MTS-TFEB) which strongly enriched TFEB in mitochondria (Fig S4B-E).

We have now performed multiple complementation experiments with the TFEB mutants which further consolidate the role of TFEB in mitochondria:

MTS-TFEB regulates mitochondrial function by modulating mitochondrial respiration:

Seahorse assay: Maximal respiration and ATP-linked respiration could be rescued in shTFEB cells when complemented with WT-TFEB and MTS-TFEB further demonstrating the mitochondria specific function of TFEB. Both Nuclear and Cytosol-localized TFEB, rescued at a less extent both the ATP-linked respiration and Maximal respiration, as they fail to translocate into the mitochondria. These data confirm that the mitochondrial TFEB regulates OXPHOS (Fig S4 F-H).

Transcriptional role: When shTFEB HeLa cells were complemented with MTS-TFEB, the transcriptional function of TFEB was not restored. Suggesting that the mitochondrial function of TFEB is distinct from its transcriptional role in the nucleus. We have included this data in Fig 4F.

Inflammation: Furthermore, MTS-TFEB was also able to restore Salmonella Typhimurium-induced inflammatory cytokines similar to WT-TFEB. However, cytosol and nucleus localized TFEB were not able to restore the inflammatory cytokines. These data clearly demonstrate that the observed mitochondrial function and the dependent-cytokine induction are primarily dependent on mitochondrial localized TFEB. We incorporated this point in Fig.7E.

4. The shown Seahorse experiments are of poor quality.

We have repeated the Seahorse experiments to substantially improve the quality.

The color code of the two subfigures should be adjusted.

We understand that the referee refers to matching colour codes. We have now matched the colour code.

5. The in-gel activity stain does not allow quantitative assessment. The author should provide additional data to show the activity of the respiratory chain complexes. The blue native gel in Figure 5 is of poor quality and should be repeated.

We have now performed the complex-I activity using commercially available colorimetry assay (Fig 5D), which confirms our observation through the in-gel activity assay.

6. The authors should show negative controls in the co-immunoprecipitations in Figure 6.

We have used IgG as a negative control and show that neither TFEB nor LonP1 interacts non-specifically with the beads or the IgG control We had also probed for mitochondrial proteins such as TFAM and we did not see any non-specific interactions (data not shown).

In the experiment where we co-immunoprecipitated with TFEB-Flag, we have used multiple controls. 1) Empty flag vector, 2) cells expressing only LONP-1-HA, 3) cells expressing only TFEB-FLAG and cells expressing TFEB-FLAG and LONP-1 HA.

We could identify LONP-1 only when TFEB-FLAG and LONP-1 HA were co-expressed.

7. The submitochondrial localization of TFEB is not entire clear. Why does TFEB interacts with TOMM20 (Figure 3I) when it localizes to the matrix? Furthermore, the localization of TFEB close to mtDNA is not convincing. These points should to be clarified.

As discussed in the manuscript, the MLS is predicted to be a TOMM20 binding motif and we have observed TOMM20 interaction upon inhibiting mTOR (Fig.3L), which actively enables translocation of TFEB into mitochondria via TOMM20. Our data suggests that TFEB translocation upon mTOR inhibition with torin-1 is dependent on TOMM20 interaction.

Since TFEB is a transcriptional factor, we wanted to investigate whether it would bind to mtDNA, therefore we performed the staining and used a high-resolution STED microscopy to verify this hypothesis. Our data shows that TFEB, although in close proximity with the mtDNA, does not directly interact with it. This data is also strengthened by the fact that depleting TFEB in HeLa cells did not result in reduced mRNA expression of mtDNA encoded protein mtND6 (Fig.S4I). Co-staining for mtDNA and TOMM20 also shows that the DNA stain specifically stains mtDNA.

We intend to show that TFEB is in the same compartment as that of the mtDNA. However, we do not conclude that TFEB could be interacting with mtDNA. We have added more clarification in the manuscript now. In addition, we have provided other evidence to show that TFEB is located in the mitochondrial matrix.

8. The authors should add a quantification to the localization of TFEB in Figure 3B. Quantification is provided in Fig 3C now.

9. Figure 3C: Import of wild-type TFEB should be included as control.

We always performed the import assay with WT-TFEB and MLS-TFEB at the same time. We had kept the data separately in 2 figures for the ease of discussing our data. We have now, modified and provided the import data for WT-TFEB and MLS-TFEB together. If the reviewer insists, we are happy to provide additional figure.

10. 3H. The authors state: "Ser467 was the only phosphorylated residue in the mitochondrial pool.... The phosphorylation of TFEB in these Ser residues was lost upon Torin treatment." The data presented here show the opposite effect. We apologise to have not stated with clarity. We have now rewritten this section.

"This analysis revealed that the pool of TFEB present in the mitochondria was not phosphorylated on the identified phosphorylation sites coloured in red (Ser122, Ser332 and Ser334) (Figure 3K). The phosphorylation of TFEB on these Ser residues was lost upon torin-1 treatment (Figure 3K). However, Ser467 was the only phosphorylated residue to be detected in the mitochondria pool upon torin-1 treatment. Ser467 phosphorylation is known to be responsible for TFEB cytosolic stabilization, as it prevents TFEB nuclear translocation independently of mTORC1 (Palmieri et al, 2017)".

11. The manuscript has to be checked in detail. E.g. first sentence of abstract which is mixed up, . mTOR is "mammalian" not "mechanistic" target of rapamycin, the terms "expression" and "steady state level" are often mixed.

mTOR has also been described as the mechanistic target of rapamycin by several prominent scientists working on mTOR (PMID: 28283069). However, we will use the reviewer's suggestion mammalian target of rapamycin. We have now carefully gone through the manuscript.

Referee #2:

In this manuscript, the authors discuss the role of TFEB, a master regulator of autophagy, lysosome biogenesis, and mitochondrial metabolism, in regulating the electron transport chain complex I in mitochondria. The study shows that TFEB has a non-transcriptional role in down-modulating inflammation through its interactions with several mitochondrial proteins. The localization of TFEB in the mitochondrial matrix was observed, and it was found that TFEB and protease LONP1 co-regulate complex I, reactive oxygen species, and the inflammatory response. Lack of TFEB specifically in the mitochondria during infection leads to the worsening of pro-inflammatory cytokine expression and contributes to innate immune pathogenesis. Although the observed mitochondrial localization of some TFEB proteins is interesting, the contradicting results observed in this study with previous literature regarding the role of TFEB in mitochondrial function need to be addressed and validated in in vivo models. The mechanism of TFEB's non-transcriptional regulation of mitochondrial respiration should also be investigated. Additionally, the authors need to exclude the possibility of mitophagy impairment upon TFEB depletion, and critical experiments should be replicated in full KO for TFEB. Finally, the localization and role of other MITF/TFE family transcription factors, such as MITF and TFE3, within the mitochondria need to be explored. Overall, the manuscript needs to address these concerns and provide more evidence to support its claims.

We are pleased to note that the reviewer is interested in our findings. The reviewer has suggested that we validate the role of TFEB in mitochondrial function in in vivo models. We thank the reviewer for the. However, we think this is beyond the scope of this manuscript and we plan to subsequently conduct these experiments. To precisely validate the role of TFEB in animal models, we will have to create a MLS-TFEB Knock In mice, which is currently not available. TFEB KO mice itself is embryonically lethal and any experiment can only be conducted in inducible knock out mice which are not commonly available. Therefore, we humbly request the reviewer to consider this point. Additionally, we have addressed this issue with the editor.

Reviewer has also suggested to exclude mitophagy. PINK1 accumulates on the outer mitochondrial membrane when mitochondrial homeostasis is altered, thus used as a hallmark of mitophagy. We see increased PINK-1 colocalization with TOMM20 in TFEB depleted cells, which we have now included in the supplementary data (Fig S5F and S5G). Furthermore, mitophagy is dependent on canonical autophagy machinery. Our data suggests that LC3 lipidation is reduced, thus autophagy is impaired upon TFEB depletion, which is expected and has been shown by several

studies. Hence, we confirm the previous findings that TFEB regulates mitophagy. However, in this manuscript, we show convincing evidence on the specific mitochondria localised function of TFEB, which is important to regulate mitochondrial homeostasis. Further, extensive experiments are required to establish the link between mitoTFEB function and mitophagy, which we believe that the reviewer can understand is beyond the scope of this manuscript.

Reviewer has also suggested to investigate the MITF and TFE3 transcription factors. In our experiments we did not observe MITF and TFE3 to be present in the mitochondrial fractions, which we have now included in the supplementary data (Fig S1G).

Other important comments and concerns:

Regarding figures in general the authors should plot the graphs showing individual replicates, not the bars. Regarding the legends, authors often do not write which cell line they use for the experiments.

We have now plotted graphs showing individual replicates as the reviewer has recommended. We have now included the name of cell lines.

Figure 1 D: I think IF of endogenous TFEB is very weak like a simple background
We have provided a better figure now.

Figure 1E: the scale should be reduced to 0,4. There are no statistics.

In Fig 1E we show the colocalization coefficient of TFEB to TOMM20 (tM1) and TOMM20 to TFEB (tM2). As we are not comparing two different groups, in this specific figure we did not include statistics. The colocalization coefficient of tM1 and tM2 is largely similar which is a validation of the colocalization. Therefore, we have not provided statistics. However, we have now reduced the scale as the reviewer has suggested. We have followed this well cited publication on the best methods for statistics used for colocalization experiments (PMID: 21209361) in which statistics are recommended only when comparing two different treated groups, which we did in Fig.S3E when comparing the tM between the WT-TFEB vs the mutant MLS-TFEB.

Figure 1F, H: Asking triplicates with densitometry and statistics

Statistics and replicates for Fig1F are shown in Fig 1G. Fig 1H depicts a representative Western blot of multiple repeats for mitochondrial enrichment in which TFEB is present. From the blot it is vivid that the mitochondrial enrichment is free of contamination from other organelles. Densitometry was not possible as we could not detect any band in the mitochondrial sample for most of the proteins tested.

Legend to Figure 1 F-H: cell line is not indicated

We have indicated the cell line in the figure legend now.

Figure 2B: WB in triplicate with densitometry and statistics

Densitometry analysis and statistics are shown in Fig 2C

Figure 2C: there are no statistics

Statistics included now.

Figure 2D: low-resolution image.

We could not get a better resolution of the EM image. The image was acquired with a resolution of 2046x2049 pixels. We are happy to provide the original picture.

Figure 2G: graph in dots of replicates, instead of bars, to add statistics

This graph has been moved to Fig 3E and as suggested we have redone the graph and conducted 1 way ANOVA statistical tests.

Figure 3A: leave only one graphical representation, instead of two

As suggested, we have kept one graphical representation now.

Figure 3B: WB in triplicate with densitometry and statistics

Western blots were repeated atleast three times, the densitometric analysis were done and statistics provided.

Figure 3E: it might be 1-way ANOVA (or 1 sample t-test)

We have done 1-way ANOVA as suggested.

Figure 3F: they can add also a confocal image of TFEB in mitochondria under Totin1 treatment in comparison to untreated cells.

We have performed confocal microscopy on cells stained for TFEB and mitochondria upon torin treatment which shows increased colocalization of TFEB with TOMM20 upon torin-1 treatment. This data is included now (Figure 3I and 3J). .

Figure 4B: Not clear the differences in morphology, no quantification, and stats

Quantification of perinuclear vs cytosolic localisation of mitochondria has been provided in Fig.4B.

Figure 4C: to add statistics .

The statistics are provided in the Fig 4D. The data used to generate the OxPhos graph was used to calculate ATP linked respiration and Maximal Respiration. Hence, we have provided statistical significance in Fig 4D.

Figure 4D: dot-plot instead of bars on the graph. Might be 1 way ANOVA (or 1 sample t-test)

We have now made graphs as the reviewer has suggested and performed 1 way ANOVA statistical test.

Figure 6B: Toa change dots instead of bars on the graph. Might be 1 way ANOVA (or 1 sample t-test)

We have changed as the reviewer has suggested and performed 1 way ANOVA.

Figure 6D: To change dots instead of bars on the graph.

We have changed as the reviewer has suggested and performed 1 way ANOVA.

Figure 6E: to add statistics to OCR

Statistics is shown in Fig 6F. The data used to generate the OxPhos graph was used to calculate ATP linked respiration and Maximal Respiration. Hence, we have provided statistical significance in Fig 6F.

Figure 6F: To change dots instead of bars on the graph. Might be 1 way ANOVA (or 1 sample t-test)

We have changed as the reviewer has suggested and performed 1 way ANOVA.

Legend to Figure 6C, F, I: cell line is not written

We have now included the cell line

Figure 7 C-E: To change dots instead of bars on the graph.

We have changed as the reviewer has suggested

Figure 7E-I: Might be 1-way ANOVA (or 1 sample t-test)

We have changed as the reviewer has suggested and performed 1 way ANOVA.

Legend to figure 7F-I: cell line is not indicated.

Cell lines are indicated now.

- Abbreviation to TFEB, ROS, mTOR, but not too many other proteins, like Optn, STAT3, etc.

We have now expanded the abbreviations.

- In the Abstract the 1st sentence is strange: "that works, and immunity, primarily through" and immunity is placed not in logical way.

In the introduction, line 54, they should cite the new link about TFEB and Rags interaction obtained from crystallography data from the recent Nature paper.

Introduction Line 75: PMID is inserted by mistake

In the Results Line 149: mistake in the name of cell line: HEK293T instead of HEK293.

We thank the reviewer to have carefully gone through the manuscript. We have now rectified these errors.

Dear Dr. Robinson

Thank you for the submission of your revised manuscript to EMBO reports. As my colleague Ioannis Papaioannou has meanwhile moved to The EMBO Journal, I have taken over the handling of your manuscript. We have now received the full set of referee reports that is copied below.

As you will see, both referees conclude that your study has been significantly strengthened during the revision but they also raise some remaining concerns. Since there is indeed a strong link between TFEB and mitophagy and since you have all mutant constructs at hand, I strongly encourage you to perform mitophagy experiments in cells reconstituted with different TFEB mutants to test whether the mitochondrial or cytosolic pool of TFEB reduces mitophagy in TFEB deplete cells.

From the editorial side, there are also a few things that we need before we can proceed with the official acceptance of your study.

- Please group the source data for the Appendix figures into one folder and then upload this as a zip file.
- Dataset EV1 is missing its legend in the Excel file - it should be added as a new sheet in the same file; the Dataset and table EV legends should be removed from the manuscript file.
- I noticed a few instances where statistical significance was determined using technical replicates only. Please see my comments in the attached manuscript file.
- I have a few comments regarding the Appendix figure legends. As I could not comment in the pdf, please find the comments below. I commented on annotating the indicated p-values per figure panel. In some cases you define the annotation or statistical test used at the end of a figure legend (e.g. S4). If this information applies to all panels showing quantification, please use: Data information: in (E) and (H - I) unpaired t-test was done....
- Appendix Fig. S1B: please define the whiskers etc shown in the box plot, the number of replicates (biological, technical).
- Appendix Fig. S2C: please change "... protein levels in Supplementary Fig. 2B" to "... protein levels in (B)". Please define the bars, the error bars, and the number of replicates (technical, biological).
- Appendix Fig. S3B: please correct "shown in Figure S3A" to "shown in (A)" and define the number of replicates (technical, biological).
- Appendix Fig. S3D: please define the error bars, the number of replicates (technical, biological) and the annotated p-values.
- S4A: please define the statistical test used and the annotated p-values.
- S4G, H: Please merge the description and use (G, H) ATP-linked respiration and (H) maximal respiration normalized to basal respiration Please define the annotated p-values.
- S5B, C: please define the whiskers etc for the box plot, the number of replicates (technical, biological), the statistical test used and the annotated p-values.
- S5E: please define the error bars and remove the statistical analysis if based on technical replicates.
- S5F: please define the size of the scale bar
- S5G: please define the nature of the replicates (technical, biological) and consider pseudoreplication in case all measurements were done on several cells from the same experiment
- S6B, C: please do not determine statistical significance based on technical replicates.
- S6D: please define the annotated p-values.
- S7B: please do not calculate statistical significance based on technical replicates or show the quantification across the independent replicates instead.
- S7C, E-H: please define the annotated p-values.
- Finally, EMBO Reports papers are accompanied online by A) a short (1-2 sentences) summary of the findings and their significance, B) 2-3 bullet points highlighting key results and C) a synopsis image that is 550x300-600 pixels large (width x height) in PNG for JPG format. You can either show a model or key data in the synopsis image. Please note that the size is rather small and that text needs to be readable at the final size. Please send us this information along with the revised manuscript.

- On a different note, I would like to alert you that EMBO Press offers a new format for a video-synopsis of work published with us, which essentially is a short, author-generated film explaining the core findings in hand drawings, and, as we believe, can be very useful to increase visibility of the work. This has proven to offer a nice opportunity for exposure i.p. for the first author(s) of the study. Please see the following link for representative examples and their integration into the article web page:

https://www.embopress.org/video_synopses
<https://www.embopress.org/doi/full/10.15252/emboj.2019103932>

Please let me know, should you be interested to engage in commissioning a similar video synopsis for your work. According

operation instructions are available and intuitive.
We look forward to seeing a final version of your manuscript as soon as possible.

With kind regards,

Referee #1:

The revised version of the manuscript improved significantly. The authors addressed several concerns of the reviewer and provide data that TFEB localizes to the mitochondrial matrix or inner membrane. The function of TFEB in mitochondria remains unclear. The new Figure S4I indicates that TFEB affects transcript levels of mtND6. What about the transcript levels of other mitochondrial genes of other complexes?

Minor points:

I suggest to move Fig. S1A to the main figure 1.

The BN-gel in Figure 5B is of bad quality.

Figure 6H: Negative controls (mitochondrial proteins that are not co-purified) for the pulldowns are missing.

Referee #2:

I appreciate the effort of the authors to improve the quality and the statistics of the manuscript. I also understand that test their claims in vivo will require a long investment on time. However, the authors did not respond to my request to analyse the link of their findings with mitophagy. I am not agree that this request is out of the scope, taking into account that the mitochondrial function of TFEB is the regulation of mitochondrial respiration upon specific stimuli and that alteration of such function can be linked to mitophagy activation. The authors have generated multiple complementation experiments with the TFEB mutants which might further consolidate the relationship between the role of TFEB in mitophagy and its mitochondrial localisation. I think that this might considerably increase the general interest of this manuscript.

As you will see, both referees conclude that your study has been significantly strengthened during the revision but they also raise some remaining concerns. Since there is indeed a strong link between TFEB and mitophagy and since you have all mutant constructs at hand, I strongly encourage you to perform mitophagy experiments in cells reconstituted with different TFEB mutants to test whether the mitochondrial or cytosolic pool of TFEB reduces mitophagy in TFEB depleted cells.

We greatly appreciate the positive feedback from the reviewers. We have now induced mitophagy by treating TFEB-depleted cells with FCCP and complemented with WT and mutant TFEB constructs. We observed that PINK-1 recruitment to mitochondria was facilitated in cells complemented with constructs that can localise TFEB in nucleus (WT and MLS). However, PINK-1 localisation with mitochondria was greatly diminished in cells complemented with constructs that localise TFEB in mitochondria, suggesting that mitochondrial localised TFEB is not involved in regulating mitophagy.

Referee #1:

The revised version of the manuscript improved significantly. The authors addressed several concerns of the reviewer and provide data that TFEB localizes to the mitochondrial matrix or inner membrane. The function of TFEB in mitochondria remains unclear. The new Figure S4I indicates that TFEB affects transcript levels of mtND6. What about the transcript levels of other mitochondrial genes of other complexes?

We thank the reviewer for the suggestion. We have now presented qPCR data for SDHA (complex II) and we do not see significant difference in transcript levels. The mRNA levels of mtND6 are also not significantly different between control, shTFEB and the complemented samples.

Minor points:

I suggest to move Fig. S1A to the main figure 1.

We have now moved it to Figure 1 as suggested.

The BN-gel in Figure 5B is of bad quality.

We have tried our best to repeat the experiments multiple times but we could not improve it further. However, we believe that the presented image clearly represents our finding. Furthermore, we have performed the complex-I activity using commercially available colorimetry assay (Fig 5D), which confirms our observation through the in-gel activity assay.

Figure 6H: Negative controls (mitochondrial proteins that are not co-purified) for the pulldowns are missing.

We did not include mitochondrial proteins that were not co-purified as we did not detect other mitochondrial proteins such as TFAM.

Referee #2:

I appreciate the effort of the authors to improve the quality and the statistics of the manuscript. I also understand that test their claims in vivo will require a long investment on time. However, the authors did not respond to my request to analyse the link of their findings with mitophagy. I am not agree that this request is out of the scope, taking into account that the mitochondrial function of TFEB is the regulation of mitochondrial respiration upon specific stimuli and that alteration of such function can be linked to mitophagy activation. The authors have generated multiple complementation experiments with the TFEB mutants which might further consolidate the relationship between the role of TFEB in mitophagy and its mitochondrial localisation. I think that this might considerably increase the general interest of this manuscript.

We have now considered the reviewer's comment on mitophagy and have performed the following. We induced mitophagy by treating TFEB-depleted cells with FCCP and complemented with WT and mutant TFEB constructs. We observed that PINK-1 recruitment to mitochondria was facilitated in cells complemented with constructs that can localise TFEB in nucleus (WT and MLS). However, PINK-1 localisation with mitochondria was greatly diminished in cells complemented with constructs that localise TFEB in mitochondria, suggesting that mitochondrial localised TFEB is not involved in regulating mitophagy.

From the editorial side, there are also a few things that we need before we can proceed with the official acceptance of your study.

- Please group the source data for the Appendix figures into one folder and then upload this as a zip file.

It has been done as recommended.

- Dataset EV1 is missing its legend in the Excel file - it should be added as a new sheet in the same file; the Dataset and table EV legends should be removed from the manuscript file.

Legend has been included in the Excel file as suggested.

- I noticed a few instances where statistical significance was determined using technical replicates only. Please see my comments in the attached manuscript file.

It has been an error on our part that it was not made clear that we have presented biological replicates and the errors have been rectified. We apologise for the error. Please see the file with the revisions marked.

- I have a few comments regarding the Appendix figure legends. As I could not comment in the pdf, please find the comments below. I commented on annotating the indicated p-values per figure panel. In some cases you define the annotation or statistical test used at the end of a figure legend (e.g. S4). If this information applies to all panels showing quantification, please use: Data information: in (E) and (H - I) unpaired t-test was done....

- Appendix Fig. S1B: please define the whiskers etc shown in the box plot, the number of replicates (biological, technical).

Done

- Appendix Fig. S2C: please change "... protein levels in Supplementary Fig. 2B" to "... protein levels in (B)". Please define the bars, the error bars, and the number of replicates (technical, biological).

Done

- Appendix Fig. S3B: please correct "shown in Figure S3A" to "shown in (A)" and define the number of replicates (technical, biological).

Done

- Appendix Fig. S3D: please define the error bars, the number of replicates (technical, biological) and the annotated p-values.

Done

- S4A: please define the statistical test used and the annotated p-values.

Done

- S4G, H: Please merge the description and use (G, H) ATP-linked respiration and (H) maximal respiration normalized to basal respiration Please define the annotated p-values.

Done

- S5B, C: please define the whiskers etc for the box plot, the number of replicates (technical, biological), the statistical test used and the annotated p-values.

Done

- S5E: please define the error bars and remove the statistical analysis if based on technical replicates.

Done

- S5F: please define the size of the scale bar

Done (Figure revised)

- S5G: please define the nature of the replicates (technical, biological) and consider pseudoreplication in case all measurements were done on several cells from the same experiment

Done

- S6B, C: please do not determine statistical significance based on technical replicates.

Done

- S6D: please define the annotated p-values.

Done

- S7B: please do not calculate statistical significance based on technical replicates or show the quantification across the independent replicates instead.

Data presented is from biological replicates. Multiple segments/slides from each replicate was analysed to arrive at the data.

- S7C, E-H: please define the annotated p-values.

Done

Dr. Nirmal Robinson
University of South Australia
Centre for Cancer Biology
Australia

Dear Dr. Robinson,

I am very pleased to accept your manuscript for publication in the next available issue of EMBO reports. Thank you for your contribution to our journal.

Yours sincerely,
